# Multifunctional cationic nanosystems for nucleic acid therapy of thoracic aortic dissection

Chen Xu[1,5], Yanzhenzi Zhang[2,5], Ke Xu[2], Jing-Jun Nie[1], Bingran Yu ◉ [1], Sijin Li[3], Gang Cheng[4], Yulin Li[2], Jie Du[2] & Fu-Jian Xu[1]

Thoracic aortic dissection (TAD) is an aggressive vascular disease that requires early diagnosis and effective treatment. However, due to the particular vascular structure and narrowness of lesion location, there are no effective drug delivery systems for the therapy of TAD. Here, we report a multifunctional delivery nanosystem (TP-Gd/miRNA-ColIV) composed of gadolinium-chelated tannic acid (TA), low-toxic cationic PGEA (ethanolamine-aminated poly(glycidyl methacrylate)) and type IV collagen targeted peptide (ColIV) for targeted nucleic acid therapy, early diagnosis and noninvasive monitoring of TAD. Such targeted therapy with miR-145 exhibits impressive performances in stabilizing the vascular structures and preventing the deterioration of TAD. After the treatment with TP-Gd/miR-145-ColIV, nearly no dissection occurs in the thoracic aortic arches of the mice with TAD model. Moreover, TP-Gd/miRNA-ColIV also demonstrates good magnetic resonance imaging (MRI) ability and can be used to noninvasively monitor the development conditions of TAD.

[1] State Key Laboratory of Chemical Resource Engineering, Key Laboratory of Carbon Fiber and Functional Polymers (Beijing University of Chemical Technology), Ministry of Education, Beijing Laboratory of Biomedical Materials, Beijing Advanced Innovation Center for Soft Matter Science and Engineering, Beijing University of Chemical Technology, Beijing 100029, China. [2] Key Laboratory of Remodeling-Related Cardiovascular Diseases (Ministry of Education), and Beijing Institute of Heart, Lung, and Blood Vessel Diseases, Beijing Anzhen Hospital Affiliated to Capital Medical University, Beijing 100029, China. [3] Department of Nuclear Medicine, The First Hospital of Shanxi Medical University, Molecular Imaging Precision Medical Collaborative Innovation Center, Shanxi Medical University, Shanxi 030001, China. [4] Department of Chemical Engineering, University of Illinois at Chicago, Chicago, IL 60607, USA. [5] These authors contributed equally: Chen Xu, Yanzhenzi Zhang. Correspondence and requests for materials should be addressed to Y.L. (email: lyllyl_1111@163.com) or to J.D. (email: jiedu@ccmu.edu.cn) or to F.-J.X. (email: xufj@mail.buct.edu.cn)

Thoracic aortic dissection (TAD) is one of the most devastating diseases[1–3]. The annual incidence recently was about 12 cases per 100,000 per year and has increased 5% per year because of aging populations and increasing incidence of hypertension[4,5]. Normally, pre-TAD was without obvious clinical manifestation. The dissection deteriorates quickly and finally leads to rupture and organ ischemia. For untreated dissections, survival rate approaches only 1% during the first 48 h. Even with the associated medical management, death rate is still 50% in the first month[6–8]. Currently, no clinical treatment can be utilized to reverse this disease. Although treated with surgical managements or interventional therapies, the death rate arising from TAD is still 30% in the first year[9]. It is of crucial importance to develop effective therapy strategies of TAD.

MicroRNAs (miRNAs), a series of highly conserved small noncoding RNA molecules, have been explored in the effective treatments of different diseases, including TAD[10–15]. MiR-145, the upstream important factor for the regulation of KLF4, is very promising for the treatment of TAD[16–18]. Among vascular diseases, KLF4 is an essential transcription factor related with the phenotypic switching of smooth muscle cells (SMCs). With the promotion of miR-145, KLF4 level will be decreased and SMCs would maintain contractile phenotypes, finally benefitting the protection from TAD[19,20]. However, miR-145 cannot be directly delivered owing to its own instability and the influence of complex environment in vivo[21,22]. Powerful delivery vectors which own high transfection efficiency and low toxicity are necessary to protect nucleic acids[23,24].

Polycations, a major type of nonviral vectors, can condense negatively-charged nucleic acids into nanocomplexes, resulting in cellular delivery and intracellular release[25,26]. Various polycation-based vectors have been synthesized for different gene therapies, for example, gold standard branched polyethyleneimine (PEI)[23–31]. In particular, a series of ethanolamine (EA)-functionalized poly(glycidyl methacrylate)s (denoted by PGEAs) with rich hydroxyl groups were developed for the effective delivery of different nucleic acids[10,12,14,32,33]. However, due to the particularity of vascular structure and narrowness of lesion location, no effective delivery vectors were reported for the nucleic acid therapy of TAD[34]. Type IV collagen can be exposed in the pathological tissues of diseases, such as TAD, atherosclerosis, hepatic fibrosis and renal fibrosis[35–40]. More recently, we found that type IV collagen exposed on the vascular intima of early TAD can be used as the excellent molecular target for early diagnosis of pre-TAD, where type IV collagen targeted peptide (ColIV) can effectively target to SMCs which express collagens IV as extracellular matrix[35]. With the introduction of ColIV, nucleic acid targeted-delivery vectors might be developed for the treatment of TAD.

In this work, one multifunctional nucleic acid delivery nanosystem (TP-Gd/miRNA-ColIV), composed of gadolinium-chelated tannic acid (TA) (for magnetic resonance imaging (MRI)), low-toxic cationic PGEA (for nucleic acid delivery) and targeted peptide ColIV (for targeting TAD), was structurally designed based on atom transfer radical polymerization (ATRP) and electrostatic assembly for miR-145 therapy of TAD (Fig. 1). MiR-145 would stabilize the vascular structure and prevent deterioration of TAD. Natural TA is widely used to target cardiac tissues and serves as one component to treat cardiovascular diseases[41], and can also effectively chelate functional inorganic elements such as gadolinium through a large number of phenolic hydroxyls[42,43]. The star-like PGEA (TP-Gd) with the gadolinium-chelated TA core can be prepared using ATRP. The introduced gadolinium in the nanosystem would benefit the early diagnosis and noninvasive monitoring of pre-TAD. The feasibility of TP-Gd/miRNA-ColIV for targeted therapy, early diagnosis and

noninvasive monitoring of TAD was investigated in detail in vitro and in vivo.

## Results

**Characterization of TP-Gd/miRNA-ColIV.** As shown in Fig. 1, TA as the core can be grafted with PGEA arms via the combination of ATRP of GMA and epoxide ring-opening reaction. Bromoisobutyryl-terminated TA (TA-Br) initiator with four initiation sites was synthesized by the reaction of BIBB with four phenolic hydroxyl groups of TA. Prior to ATRP, the extra phenolic hydroxyls of TA-Br were chelated with gadolinium to produce TA-Br-Gd, which could protect the ATRP catalyst from the interferences of phenolic hydroxyls to ensure the success of ATRP. The content of chelated gadolinium accounted for nearly 20% of the total initiator (Supplementary Fig. 1). TA-PGMAs with different molecular weights can be readily synthesized through ATRP. Our earlier work indicated that PGEA derivatives from PGMA with the molecular weight of about $2 \times 10^4$ g mol$^{-1}$ exhibited fairly good gene transfection performance[44–46]. In this work, TA-PGMA of about $2.20 \times 10^4$ g mol$^{-1}$ was selected for the preparation of TA-PGEA (TP) (Supplementary Table 1).

TP was synthesized through the ring-opening reaction of epoxy groups of TA-PGMA with excess EA. Due to the loss of cheated gadolinium during the preparation process, the prepared TP was subsequently supplemented with more gadoliniums to produce TA-PGEA-Gd (TP-Gd). The mass ratio of gadolinium was increased to be 8.10 (for TP-Gd) from 0.32 for TP (Supplementary Table 1). It was difficult to directly modify TP-Gd with targeted peptide ColIV. The interaction of gadolinium in TP-Gd with peptide would lead the instability and sedimentation of TP-Gd and peptide in water. Therefore, as shown in Fig. 1 the alternative PGMA derivative with some primary amine species, TA-PGEA/ED (TP-E), was synthesized for the conjugation of ColIV. TP-E was prepared via ring-opening reaction of epoxy groups of TA-PGMA with excess EA and ethyl diamine (ED), where the molar ratio of EA and ED was about 7:1 (Supplementary Fig. 2). The primary amino groups of TP-E were used via amidation reaction to introduce ColIV and produce TP-ColIV. The typical $^1$H NMR spectra of TA-Br with four initiation sites, TA-PGMA, TP, TP-E, and TP-ColIV were shown and analyzed in detail in Supplementary Fig. S2, which indicated the successful synthesis of all the materials. Rhodamine B modified ColIV was used to evaluate the reaction efficiency of peptide via fluorescent quantitation. Approximately eight targeted peptides were introduced to one TP-E (Supplementary Fig. 3).

It is crucial for cationic vectors to effectively condense nucleic acid via electrostatic interaction and form nanocomplexes with suitable sizes and surface charges. Polycation-to-miRNA ratios were expressed as the molar ratios (or as N/P ratios) of nitrogen (N) in polycation to phosphate (P) in miRNA. The condensability of complexes were evaluated by agarose gel electrophoresis, particle size and ζ-potentials methods and atomic force microscopy (AFM) imaging. The gold-standard nonviral transfection polymer, branched polyethyleneimine (PEI, 25 kDa), was also tested as the control. As shown in Fig. 2a, TP-Gd and TP-ColIV could completely condense miRNA when the N/P ratio reached 2, similar to TP and PEI. The particle sizes of all the polycation/miRNA complexes were ranged from 100 to 300 nm (Fig. 2b) and which were suitable for cellular internalization[32,33]. With the increase of N/P ratio, more and more positive charges assisted compressing miRNA, leading to smaller and tighter nanoparticles.

In this work, electrostatic assembly was further utilized to construct the TP-Gd/miRNA-ColIV nanosystem for targeted gene therapy. TP-Gd firstly condensed miRNA into the

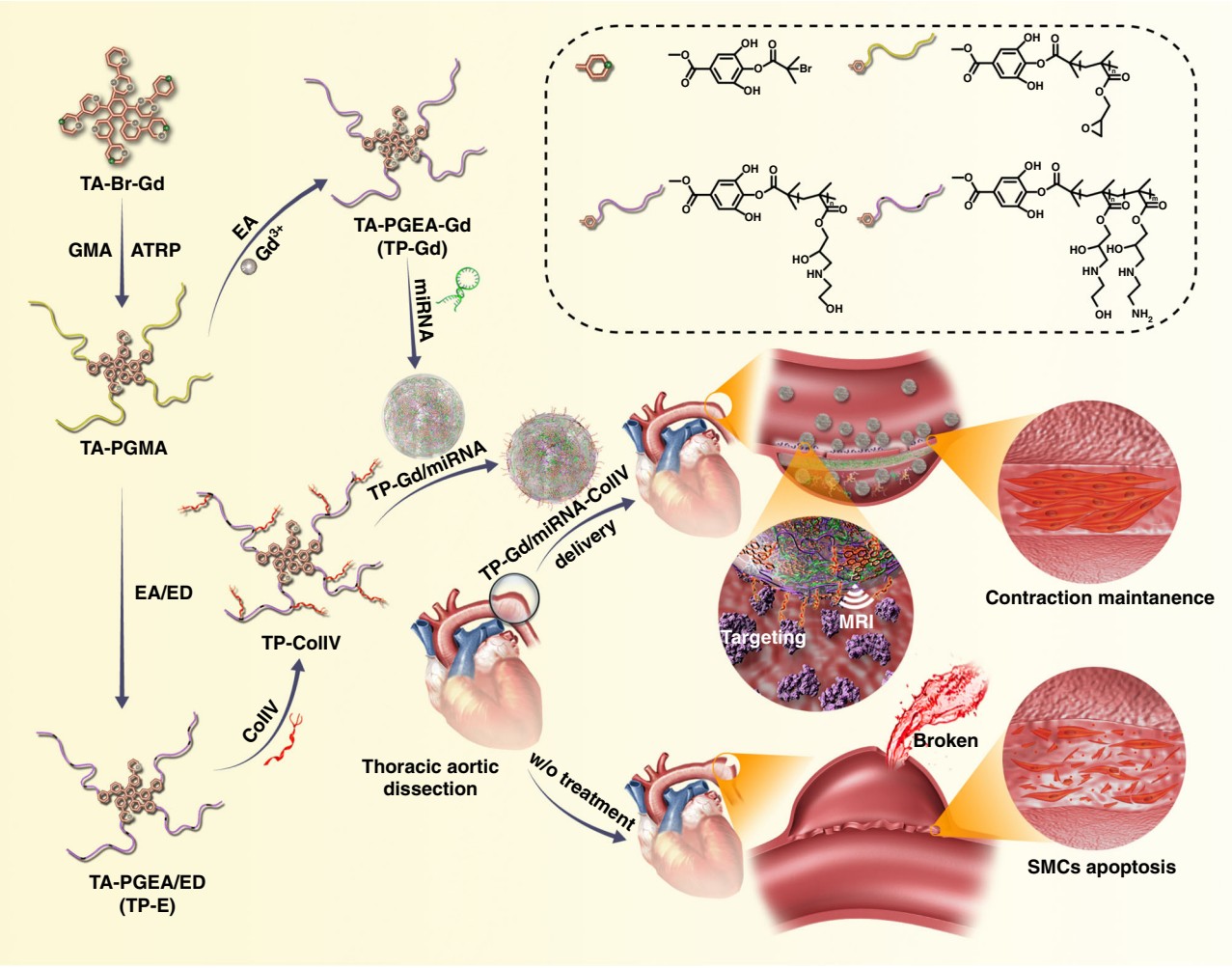

**Fig. 1** Schematic diagram illustrating the preparation of TP-Gd/miRNA-ColIV complexes and the resultant targeted gene therapy of thoracic aortic dissection (TAD)

preliminary particle (at the low N/P ratio of 5) which was then further complexed by TP-ColIV at the total N/P ratio of 10 to produce TP-Gd/miRNA-ColIV. The particle size of TP-Gd/miRNA-ColIV was mildly larger than TP-Gd/miRNA at the same N/P ratio of 10 probably because of the occupation of ColIV (Fig. 2b). ζ-potentials represent the surface charges of nanocomplexes. The ζ-potentials of all the complexes were ranged from 20 to 40 mV (Fig. 2b). The positive ζ-potentials could benefit the attraction between complexes and negatively-charged cell membranes, increasing the efficiency of cellular uptake. Meanwhile, the AFM images of different polycation/miRNA complexes at the representative N/P ratio of 10 were observed to further confirm the condensation ability and specific morphology (Fig. 2c). All the complexes exhibited relatively uniform spheres.

**Protein absorption and biocompatibility assay**. The anti-protein adsorption capacity and biocompatibility of polycations are also essential for in vivo applications. The electronegative molecules such as human serum albumin would weaken the stability of complexes. The possible severe hemolysis from polycations might result in thrombosis. Different polycation/miRNA complexes were treated with bovine serum albumin (BSA) to evaluate the anti-protein adsorption capacity (Supplementary Fig. 4). Compared with PEI/miRNA which rapidly absorbed over 70% of BSA within 0.5 min, TP-Gd/miRNA exhibited the better serum-tolerant ability and

demonstrated significantly lower protein adsorption (below 25% at all the concentrations). In addition, the sizes of TP-Gd/miRNA and TP-Gd/miRNA-ColIV were quite stable in the medium with serum within a 0–5 h incubation (Supplementary Fig. 5), providing the possibility of the long blood circulation.

Hemolysis assay was taken to evaluate the blood compatibilities. The hemolysis ratios of TP, TP-Gd, and TP-ColIV were much lower (below 5%) than that of PEI (over 15%) (Supplementary Fig. 6). Moreover, the morphologies of red blood cells (RBCs) in different groups (Fig. 2d) illustrated that PGEA-based polycations would not cause structural damage to RBCs which is opposite to PEI. Low cytotoxicity was also the focus of biomaterials. In comparison with PEI/miRNA, the cytotoxicities of TP-containing complexes were much lower at various N/P ratios in SMCs (Fig. 3a). Distinguishing form many irregular amines of gold-standard PEI, PGEA-based polycations have not only secondary amines, but also abundant hydroxyls. These plentiful hydroxyls could benefit biocompatibility[10,12]. All the above excellent abilities of TP-based materials benefited the possibility for the in vivo transfection of TP-Gd/miRNA-ColIV.

In vitro **cellular uptake and gene expression assay**. For effective gene transfection, efficient cellular internalization is also necessary. Fluorescent dye Cy3 labeled miRNA was utilized to evaluate the efficiency of cellular uptake (Fig. 3b). With the increase of N/

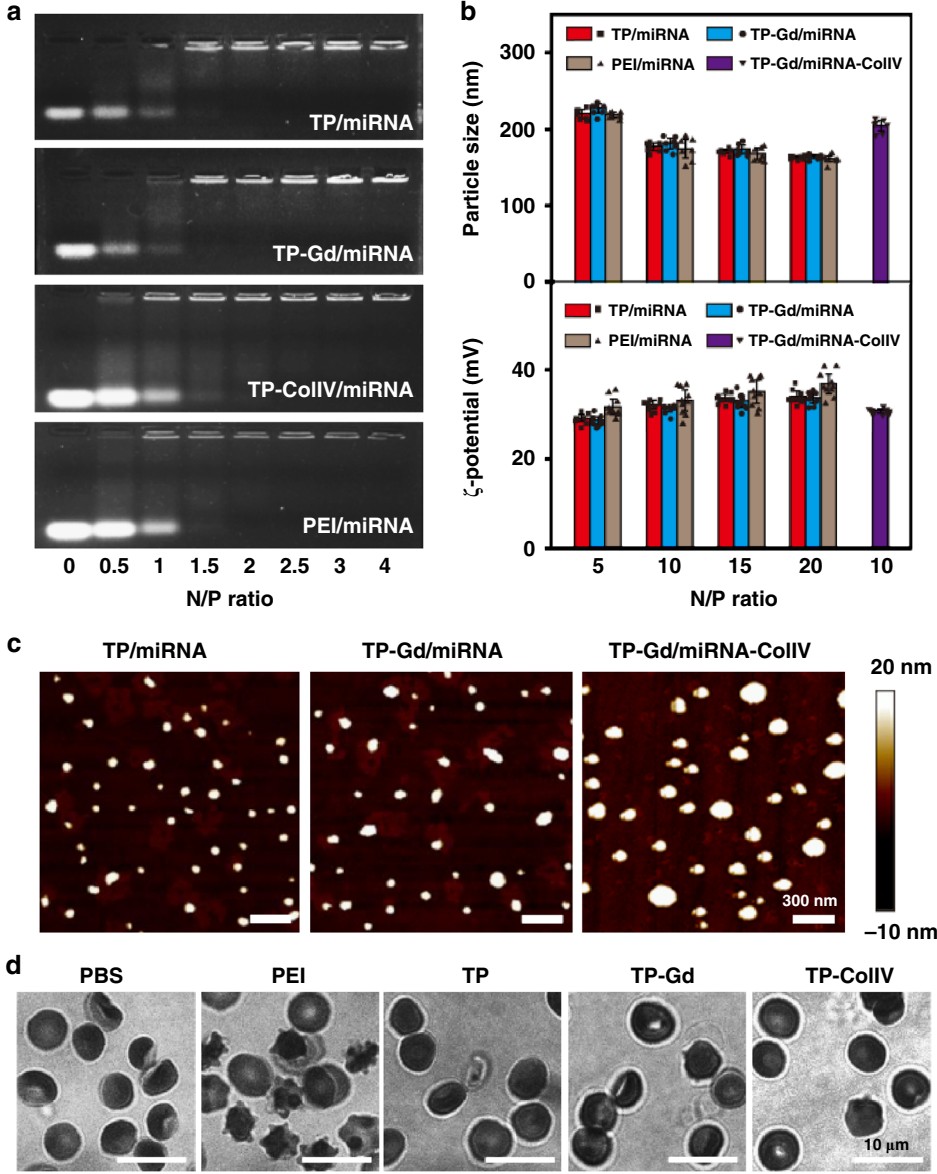

**Fig. 2 a** Electrophoretic mobility retardation assay of miRNA in the complexes with PEI, TP, TP-Gd, and TP-ColIV at various N/P ratios. **b** Particle size and ζ-potential of the polycation/miRNA complexes at various N/P ratios. **c** AFM images of different polycation/miRNA complexes at the typical N/P ratio of 10. **d** Images of red blood cells (RBCs) treated with PEI, TP, TP-Gd, and TP-ColIV at the concentration of 1 mg mL$^{-1}$, where PBS was used as the control. ($n \geq 7$ independent experiments; error bars represent standard deviation; source data of **a**, **b** are provided as a Source Data file)

P ratio, oversaturated cations will exhibit some cytotoxicity and lead to the decrease in the number of positive cells for all polycations. TP-based polycation/miRNA showed better cellular uptake capabilities than PEI/miRNA. Based on the balance between cytotoxicity and cellular internalization, the following biological performances of PEI/miRNA, TP-Gd/miRNA, and TP-Gd/miRNA-ColIV were assayed at the optimal N/P ratio of 10.

Intuitive images were also captured by confocal laser scanning microscopy (CLSM) to characterize the cellular uptake of different complexes in SMCs (Fig. 3c). Nuclei stained by DAPI emitted blue light and miR-Cy3 emitted red light. The fluorescent intensity of native miR-Cy3 in cells was very low due to the ready degradation by extracellular RNA enzyme and metabolization by intracellular lysosome. TP-Gd/miR-Cy3-ColIV and TP-Gd/miR-Cy3 demonstrated higher intensity than PEI/miR-Cy3. It was also noted that TP-Gd/miR-Cy3-ColIV exhibited the best performance. Their corresponding quantitative signal intensities also

showed the consistent results (Fig. 3d). The above data illustrated that TP-Gd/miRNA-ColIV could effectively deliver more miRNAs into SMCs. From our previous study[35], the large amount of type IV collagen secreted by SMCs could be effectively identified by targeted peptide ColIV, thus improving the endocytosis of TP-Gd/miRNA-ColIV.

The expression level of delivered miR-145 in SMCs was measured by quantitative real-time polymerase chain reaction (qRT-PCR), where U6 was set as the internal reference. In comparison with the blank and miR-neg (miRNA with no function) groups, the relative expression level of miR-145 increased (Fig. 3e). Moreover, TP-Gd/miR-145-ColIV was more effective than PEI/miR-145 and TP-Gd/miR-145 due to the enhanced endocytosis of miR-145 from the targeting effect. The expression inhibition of KLF4 could benefit for preventing aneurysm formation, and miR-145 can effectively inhibit KLF4 protein expression[16–20]. The regulation of KLF4 by delivered

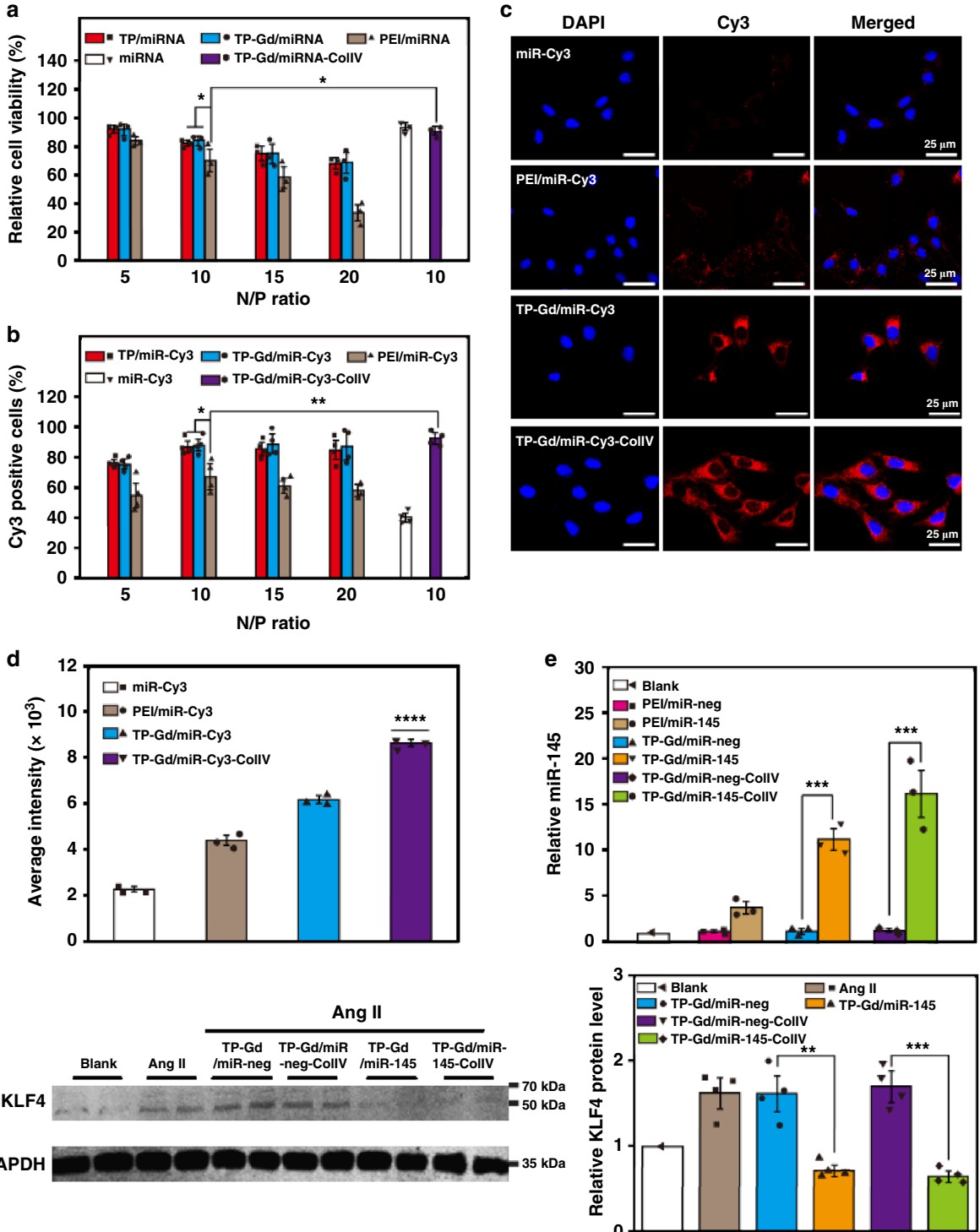

**Fig. 3 a** Cytotoxicity and **b** cellular uptake of polycation/miRNA complexes at different N/P ratios in smooth mouse cells (SMCs). Cellular internalization of polycation/miRNA complexes at the N/P ratio of 10 in SMCs: **c** Confocal fluorescence images, **d** average intensity measured by MD, **e** relative expression levels of miR-145 and **f** related KLF4 protein expressions determined by western blot. (*$p < 0.05$, **$p < 0.01$, ***$p < 0.001$ and ****$p < 0.0001$, see "Statistical analysis" in "Methods"; $n \geq 3$ independent experiments; error bars represent standard deviation; Source data of **a**, **b**, **d**–**f** are provided as a Source Data file)

miR-145 was measured by western blot (WB) (Fig. 3f). Due to the low expression of KLF4 in normal SMCs, all the experimental groups were treated with Ang II primarily, which could improve the expression of KLF4 in SMCs[47]. The significantly deeper band in the Ang II group than the blank group illustrated the successful upregulation of KLF4 in SMCs. In the miR-neg groups, the expression level of KLF4 did not decrease. In contrast, the expression amounts of KLF4 in the TP-Gd/miR-145 and TP-Gd/miR-145-ColIV groups were decreased to a third of the Ang II group. On the other hand, no obvious difference in the expression levels of KLF4 was observed between PEI/miR-neg and PEI/miR-145 (Supplementary Fig. 7). The above results confirmed that TP-Gd/miR-145-ColIV could effectively deliver miR-145 into cells and decrease the expression level of KLF4 in vitro.

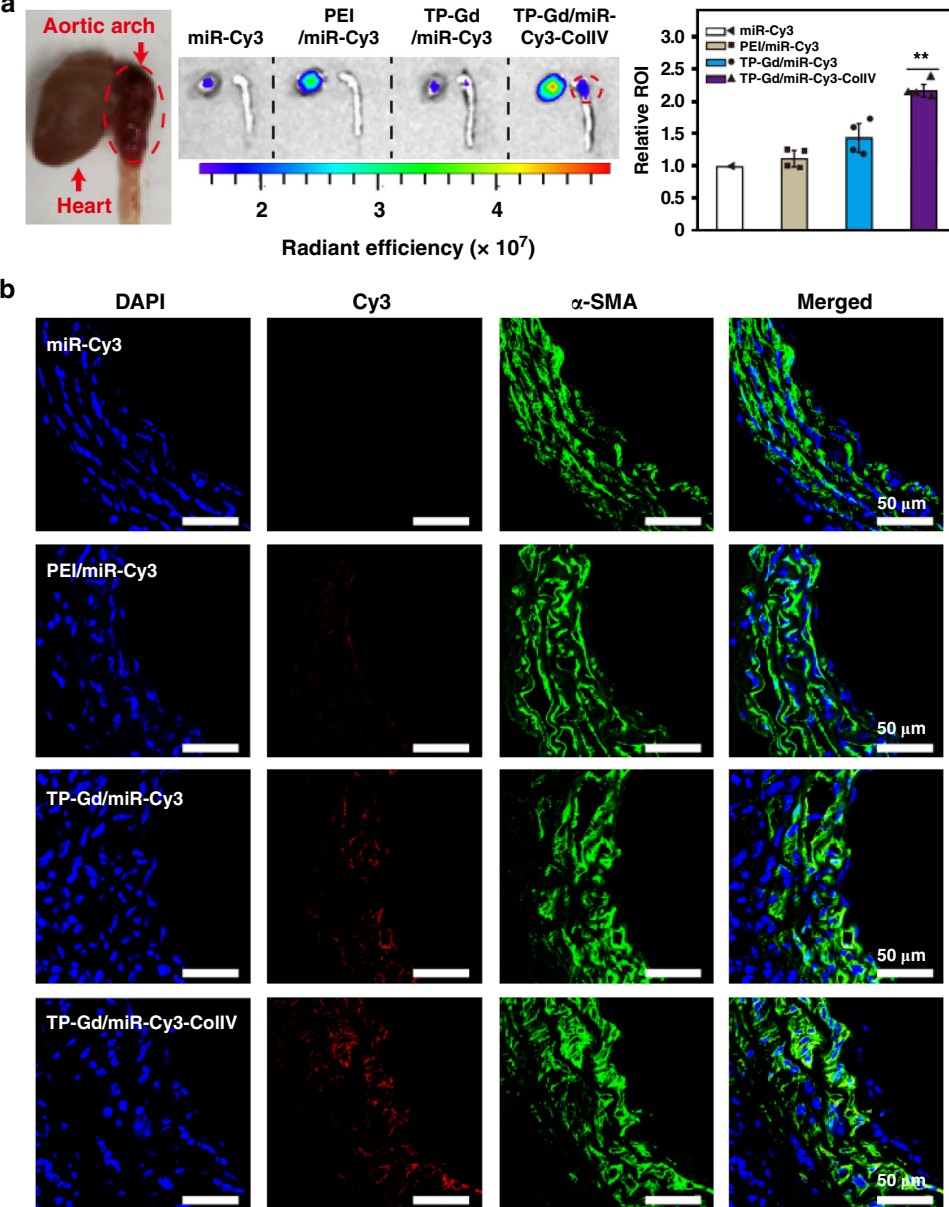

**Fig. 4 a** Representative images and relative ROI of thoracic aorta with early dissection after the accumulation of miR-Cy3 and polycation/miR-Cy3 complexes determined by Xenogen IVIS imaging system (**$p < 0.01$, see "Statistical analysis" in "Methods"; source data are provided as a Source Data file). **b** In vivo fluorescent images of accumulated miR-Cy3 in mouse thoracic aortic arch. ($n \geq 3$ animals; error bars represent standard deviation)

**In vivo cellular uptake assay**. To construct early experimental TAD models, *C57BL/6J* mice were treated with β-aminopropionitrile (BAPN, an inhibitor of lysyl oxidase) for two weeks, which could lead to the typical degradation of extracellular matrix (ECM) and the loss of SMCs[48,49]. The targeting ability of nanosystems was measured during the dissection formation, and the two-week TAD model with no obvious dissection was used for in vivo cellular uptake assay. The mice with two-week TAD model were treated with free miR-Cy3, PEI/miR-Cy3, TP-Gd/miR-Cy3, and TP-Gd/miR-Cy3-ColIV, respectively. All the treated mice were sacrificed for live imaging after different time points of 0–24 h (Fig. 4a and Supplementary Fig. 8). The precise detection position of aortic arch was also shown in Fig. 4a. Without delivery vectors, free miR-Cy3 was easily cleared by blood, and no fluorescent signal existed in thoracic aortic arch. Although PEI could in vitro protect miR-Cy3 (Fig. 3b), the fluorescent signal of PEI/miR-Cy3 was much lower than those of the TP-involving groups. That was probably because of the strong protein

adsorption capacity (Supplementary Fig. 4) and poor blood compatibility (Fig. 2d and Supplementary Fig. 6) of PEI. In comparison with PEI, the TP-Gd/miR-Cy3 and TP-Gd/miR-Cy3-ColIV groups exhibited better performance. During the preliminary formation of aortic dissection, the progressive disorganized intima of vessel probably facilitated the aggregation of complexes in thoracic aortic arch[35]. The TP-Gd/miR-Cy3-ColIV group demonstrated the strongest signal of thoracic aorta at all time points because of the targeting effect of SMC. The relative statistics of the relative region of interest (ROI) of different groups (miR-Cy3 group was used for normalization) was also shown in Fig. 4a, which exhibited the same trend. As shown in Supplementary Fig. 8, fluorescent signals of miR-Cy3 increased in livers and kidneys with increasing time, demonstrating the metabolic process.

In addition, the fluorescent images of arcus aortae sections from different groups were shown in Fig. 4b. SMCs in vessels were stained in green by immunofluorescence staining of α-SMA.

Cy3 fluorescence could hardly be detected in the free miR-Cy3 and PEI/miR-Cy3 groups. It was observed that miR-Cy3 was delivered by different TP-containing complexes to SMCs according to the location of α-SMA. The strongest fluorescent signal in the arcus aortae sections was observed in the TP-Gd/miR-Cy3-ColIV group, consistent with the results of Fig. 4a. The above data indicated that TP-Gd/miRNA-ColIV is very promising for effective in vivo nucleic acid delivery.

**Gene therapy of TAD with miR-145**. In vivo assay mediated by TP-Gd/miR-145-ColIV was subsequently performed with 4-week TAD models to take full observation of therapeutic effects. *C57BL/6J* male mice were treated with β-aminopropionitrile (BAPN) for four weeks to construct TAD models (Fig. 5a). All the mice were divided into six groups and treated via angular vein with PEI/miR-neg, PEI/miR-145, TP-Gd/miR-neg, TP-Gd/miR-neg-ColIV, TP-Gd/miR-145 or TP-Gd/miR-145-ColIV containing 2.5 nmol miRNA every three days from the second week to the fourth week. The mice fed with saline were set as blank control. The recorded survival curve was shown in Fig. 5b. The survival ratio in the blank group was 100%. In comparison with the survival ratios (<45%) of the PEI/miR-neg, TP-Gd/miR-neg, TP-Gd/miR-neg-ColIV, and PEI/miR-145 groups, the TP-Gd/miR-145 and TP-Gd/miR-145-ColIV groups demonstrated higher survival ratios. Particularly, TP-Gd/miR-145-ColIV produced the highest survival ratio (more than 80%).

Four weeks later, all the mice were sacrificed to evaluate the indicators associated with TAD. Typical anatomical thoracic aortas from different groups were shown in Fig. 5c and Supplementary Fig. 9a. No TAD occurred in healthy mice. Obvious TAD appeared in the PEI/miR-neg, TP-Gd/miR-neg, TP-Gd/miR-neg-ColIV, and PEI/miR-145 groups. Dissection was slightly inhibited in the TP-Gd/miR-145 group. More importantly, the TP-Gd/miR-145-ColIV group did not exhibit obvious dissection. The incidence of TAD was analyzed and calculated from all the anatomical thoracic aortas (Fig. 5d). Differing from the PEI/miR-145 and TP-Gd/miR-145 groups (where the incidences of TAD were more than 60%), the TP-Gd/miR-145-ColIV group (~21%) could effectively prevent the formation of TAD. Figure 5e showed the statistical diameters of thoracic aortic arch. The normal diameter of mouse in this stage was nearly 0.5 mm. The PEI/miR-145 and TP-Gd/miR-145 groups showed larger diameters. Without effective treatment, dissection would continuously expand and the corresponding diameter of TAD also sequentially increases. The diameter of thoracic aortic arch in the TP-Gd/miR-145-ColIV group maintained in the normal range (0.4~0.6 mm) due to the best therapeutic effect.

Pathological analysis of thoracic aortic arches was studied based on H&E and elastin stainings (Fig. 5f and Supplementary Fig. 9b). From H&E staining, comparing with the blank group, obvious dissections occurred in the PEI/miR-neg, TP-Gd/miR-neg and TP-Gd/miR-neg-ColIV groups, where the formed huge false lumens were in the risk of rupture. In particular, for the TP-Gd/miR-145-ColIV group, aortic wall was intact and stable. Elastin staining was also used for evaluating the integrity of vascular elastic fibers. No disorders occurred in the aortic wall of mice treated with TP-Gd/miR-145-ColIV, distinguishing from the other groups. The above data further illustrated that TP-Gd/miR-145-ColIV could effectively prevent the development of TAD.

It was also vital to determine whether the excellent therapeutic effect of TP-Gd/miR145-ColIV was based on the pathway of downregulation of KLF4. Relative miR-145 levels in thoracic aorta of different groups were shown in Fig. 6a, which confirmed that TP-Gd/miRNA-ColIV could effectively transfer miR-145 into thoracic aorta. The corresponding immunohistochemical staining of thoracic aortic arch was shown in Fig. 6b and Supplementary Fig. 10. Red arrows were used to highlight the positive area. The PEI/miR-neg, TP-Gd/miR-neg, TP-Gd/miR-neg-ColIV, and PEI/miR-145 groups exhibited apparent brown areas in thoracic aortic arch which represented the expression of KLF4. The KLF4-positive area in TP-Gd/miR-145 groups slightly decreased, while the TP-Gd/miR-145-ColIV group demonstrated small KLF4-positive areas. Accordingly, the TP-Gd/miR-145-ColIV group also demonstrated the substantially lower staining intensity (4.7%) of KLF4-positive area (Supplementary Fig. 11). The in vivo regulation of KLF4 with delivered miR-145 was further quantified by WB (Fig. 6c). Compared with other groups, the expression of KLF4 in the TP-Gd/miR-145-ColIV group was decreased to be the level of the blank group. All these data indicated the TP-Gd/miR-145-ColIV group produced the best in vivo downregulation of KLF4, which was much more obvious than the in vitro case (Fig. 3f). Targeting ability of drug delivery systems was more essential in vivo for intravenous (i.v.) injection than in vitro.

Meanwhile, α-SMA, SM22α, and *myh11* markers, which were positively correlated with contractile phenotypes of SMCs, were also detected and shown in Fig. 6a and Supplementary Fig. 12. The TP-Gd/miR-145-ColIV group showed the highest mRNA and protein levels of SM22α and myh11, further confirming the best performance of TP-Gd/miR-145-ColIV in preventing TAD.

**Early diagnosis and noninvasive monitoring assay of TAD**. Currently, clinical diagnosis of TAD relies on medical imaging methods, such as MRI, transthoracic echocardiography (TTE) and computed tomography angiography (CTA)[50,51]. However, these methods cannot provide early detection of dissection before intimal tear (the process of pre-TAD). Molecular imaging, which is the combination of early characteristic pathological processes and classic imaging, is developed for such kind of diseases[52–54]. More recently, we successfully designed one type IV collagen-targeted MRI probe for the early diagnosis of TAD[35]. In this work, the possibility of TP-Gd/miRNA-ColIV was also explored for the early diagnosis and monitoring of TAD. MRI intensities of different materials in phosphate buffer saline (PBS) (Supplementary Fig. 13) were firstly measured. The signal intensity of inverse $T_1$ was positively correlated with the concentration of $Gd^{3+}$. In comparison with TP, more gadoliniums in TP-Gd contributed to the higher relaxation efficiency (the slope of the linear fitting) and the stronger MRI ability. Due to the better cellular-uptake efficiency (Fig. 3c), TP-Gd/miR-neg-ColIV produced higher relaxation efficiency in SMCs than TP-Gd/miR-neg (Fig. 7a and Supplementary Fig. 14). On the other hand, it was noted that the MRI ability of commercial contrast MultiHance® did not increase obviously with increasing the concentration of $Gd^{3+}$ in SMCs, probably due to the low cellular internalization.

Most TP-Gd-containing complexes could be degraded in vivo within two days (Fig. 7b). It was investigated whether TP-Gd/miR-neg-ColIV could be used as an early MRI diagnostic agent of TAD (Fig. 7c and Supplementary Fig. 15). MRI plain scan (MRI scan①) was performed on the mice (which was treated with BAPN for two weeks) as the MRI background. Then, the mice were injected with TP-Gd/miR-neg and scanned (MRI scan②). After two days of degradation, these mice were plain scanned again to ensure no interference of TP-Gd/miR-neg, then treated with TP-Gd/miR-neg-ColIV and scanned (MRI scan③). As shown in Fig. 7c, the significant enhancement did not occur in the aortic arch of mice after the plain scanning and injection of TP-Gd/miR-neg. However, due to the targeting effect, the bright areas representing early TAD were clearly

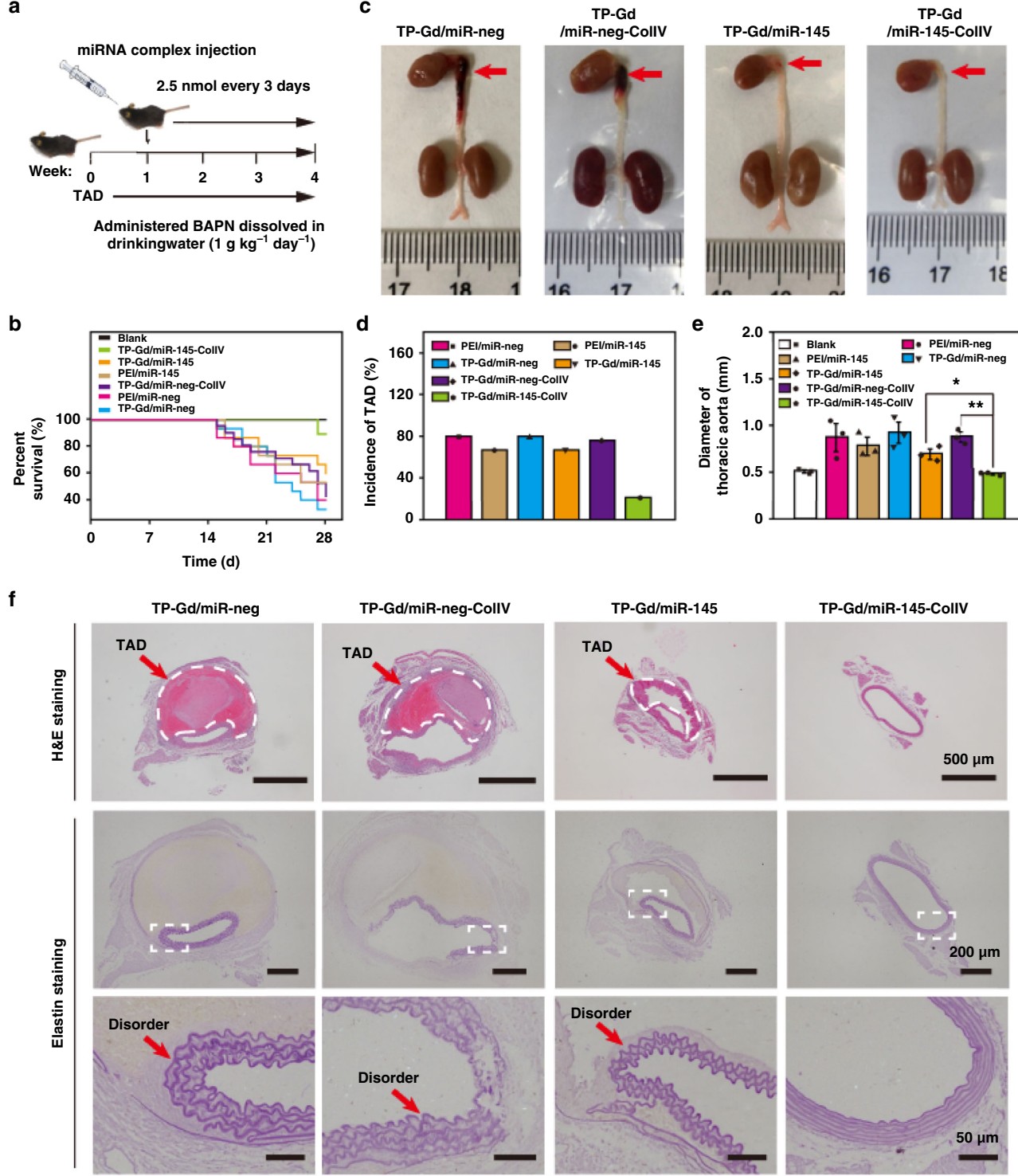

**Fig. 5** In vivo effects of saline and polycation/miR-145 complexes on BAPN-induced TAD model. **a** Time axis of administration, **b** percent survival, **c** general phenotypic pattern captured by digital camera, **d** incidence of TAD and **e** diameters of thoracic aortic arch of all the groups, and **f** representative images of H&E- and elastin-thoracic aortic arches from all the treated mice. (*$p < 0.05$ and **$p < 0.01$, see "Statistical analysis" in "Methods"; $n \geq 15$ animals; error bars represent standard deviation; source data of **b**–**d** are provided as a Source Data file)

observed after the injection of TP-Gd/miR-neg-ColIV. Also, based on the analysis of normalized percentage signal enhancement (%NSE), the enhancement efficiency of the TP-Gd/miR-neg-ColIV group could reach to nearly 30%, which was much higher than 4% of the TP-Gd/miR-neg group. The above results demonstrated that TP-Gd/miR-neg-ColIV could be used for early diagnosis of TAD.

In addition to early diagnosis, TP-Gd/miR-neg-ColIV was also used for monitoring the development of TAD. During the treatments of TP-Gd/miR-neg-ColIV and TP-Gd/miR-145-ColIV (Fig. 6a), the MR images of thoracic aortic arches (Fig. 7d and Supplementary Fig. 15) were acquired in the third week and in the fourth week by using the detection agent TP-Gd/miR-neg-ColIV. For the treatment group with control TP-Gd/miR-

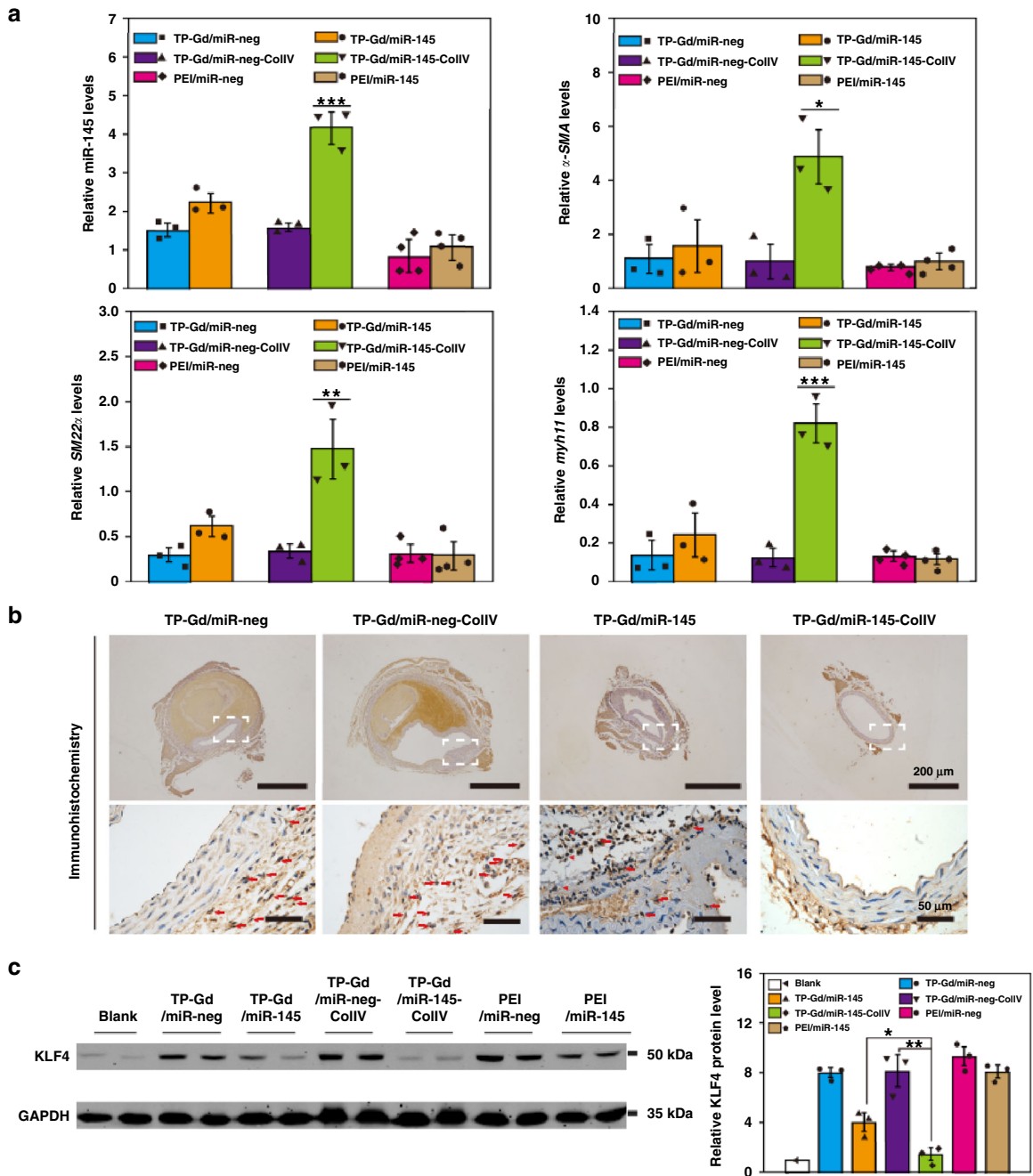

**Fig. 6 a** Relative miR-145 levels and mRNA expression of *α-SMA*, *SM22α* and *myh11* quantified by PCR assays in thoracic aorta for different groups. **b** Representative images of KLF4-positive immunohistochemical-stained thoracic aortic arches from all the treated mice (red arrows highlighted positive areas). **c** Relative KLF4 protein expressions in vivo determined by western blot. (*$p < 0.05$, **$p < 0.01$ and ***$p < 0.001$, see "Statistical analysis" in "Methods"; $n \geq 3$ independent experiments; error bars represent standard deviation; source data of **a**, **c** are provided as a Source Data file)

neg-ColIV, the lesion areas had become brighter and even the dissection could be clearly observed. On the contrary, no obvious signals were observed in the treatment group with TP-Gd/miR-145-ColIV. Also, the corresponding analysis of NSE% illustrated the same conclusion (Fig. 7d). For the TP-Gd/miR-neg-ColIV group, NSE% was gradually enhanced in the process of TAD and kept in the high levels (>50%), representing the deterioration of TAD. NSE% of the TP-Gd/miR-145-ColIV group maintained in the low levels (<10%), indicating that the development of TAD was prevented. The above monitoring results were consistent with those of the above in vivo treatment experiments (Fig. 5), illustrating that TP-Gd/miRNA-ColIV

could be used to effectively monitor the development of TAD during the treatment.

**In vivo biocompatibility assay**. In addition to in vitro cytotoxicity (Fig. 3a), it was also of crucial importance for in vivo biomedical applications to study the organ toxicity of TP-Gd/miRNA-ColIV. After the in vivo treatment of different groups as shown in Fig. 6a, the functions of heart, liver, spleen, lung, and kidney in mice of different groups were evaluated by H&E staining (Fig. 8 and Supplementary Figs. 16, 17). No obvious pathological lesions were observed in the sections of all the

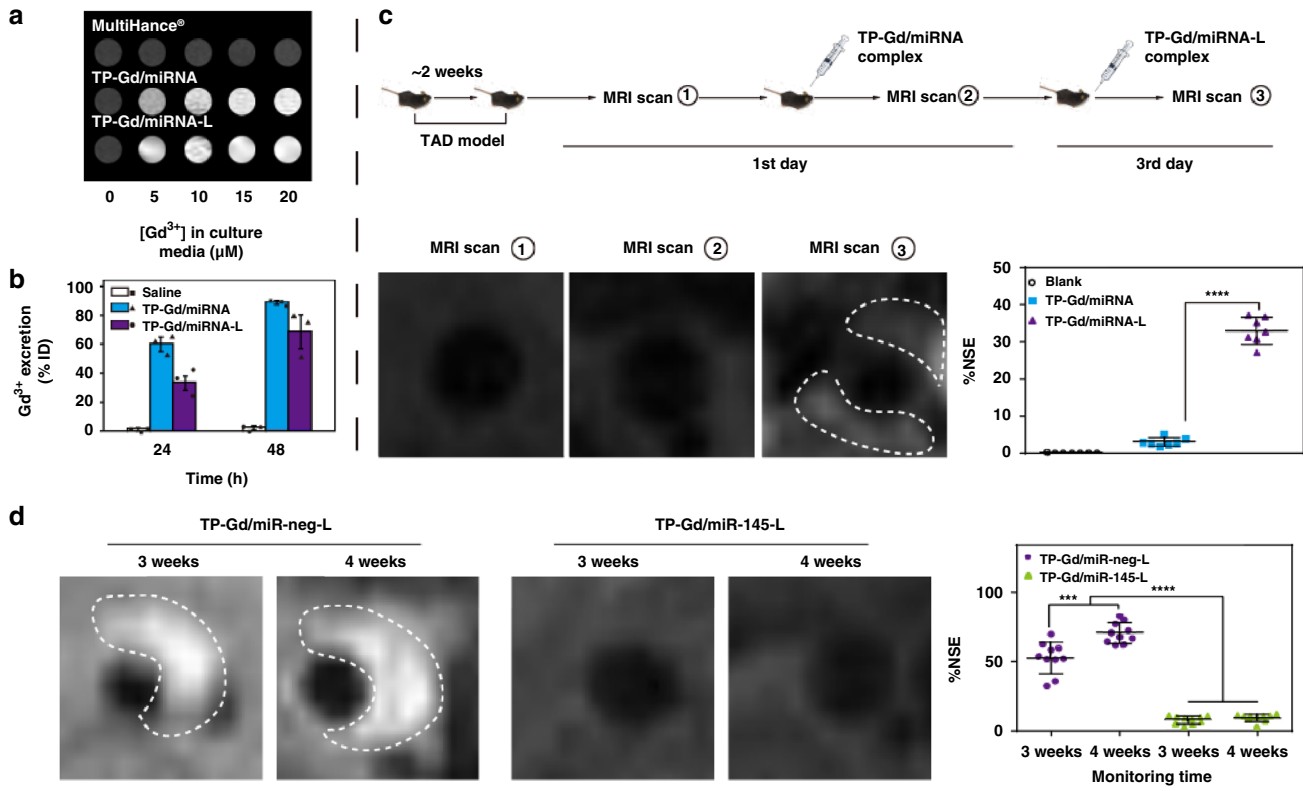

**Fig. 7 a** $T_1$-weighted MR images in SMCs under different concentrations of $Gd^{3+}$. **b** ICP-MS analysis of $Gd^{3+}$ amounts in the urine of mice collected at different periods after the infection of saline, TP-Gd/miR-neg or TP-Gd/miR-neg-ColIV. **c** MRI diagnosis and noninvasive monitoring on BAPN-induced TAD model: specific experimental procedures of diagnosis, representative MR images (white dashed area represented MRI enhancement) and normalized percentage signal enhancement (%NSE) of thoracic aortic arches from two-week TAD models after the injection of different complexes. **d** Representative MR images (white dashed area represented dissections) of TAD and %NSE of thoracic aortic arches at different periods during the treatments of TP-Gd/miR-neg-ColIV and TP-Gd/miR-145-ColIV, where TP-Gd/miR-neg-ColIV was used as the detection agent. (***$p < 0.001$ and ****$p < 0.0001$, see "Statistical analysis" in "Methods"; $n = 3$ animals, error bars represent standard deviation in Fig. 7b; $n \geq 7$ animals, error bars represent standard deviation in **c**, **d**; source data of **b–d** are provided as a Source Data file)

groups. For example, they exhibited the normal cardiac muscle tissues in the hearts and the normal hepatic lobular architectures in livers. The organ-associated toxicities were also assessed by the test of blood samples of the mice from different groups. As shown in Fig. 9, the indexes of liver function (aspartate transaminase (AST), alanine transaminase (ALT), and total bilirubin (TBIL) levels), renal function (blood urea nitrogen (BUN) and creatinine (CRE) levels) and cardiac toxicity (creatine kinase (CK) level) of all the test groups were similar to those of the blank group. Blinded histopathology scoring assay was also performed to study organ toxicity. All the above results demonstrated that TP-Gd/miRNA-ColIV would not cause damage to organs.

## Discussion

In summary, one multifunctional nucleic acid delivery nanosystem (TP-Gd/miRNA-ColIV), composed of gadolinium-chelated TA, low-toxic cationic PGEA, and targeted peptide ColIV, was successfully prepared based on ATRP and electrostatic assembly for targeted therapy, early diagnosis and noninvasive monitoring of TAD. TP-Gd/miRNA-ColIV could effectively deliver miR-145 to stabilize the vascular structure and prevent the deterioration of TAD. Due to its good MRI ability, TP-Gd/miRNA-ColIV also can be used to monitor the development of TAD. Moreover, TP-Gd/miRNA-ColIV possesses good blood compatibility and did not cause toxicity to organs. Such an excellent treatment strategy would be applied to the people who are susceptible to TAD, such

as marfan patients. Due to the wide existence of exposed type IV collagen in pathological tissues, such multifunctional miRNA delivery nanosystem would contribute valuable information for therapies of different diseases in addition to TAD.

## Methods

**Materials.** Tannic acid (TA), branched polyethyleneimine (PEI, Mw ~ 25,000 Da), 2-bromoisobutyryl bromide (BIBB, 98%), gadolinium trichloride hexahydrate (GdCl$_3$·6H$_2$O, 99%), glycidyl methacrylate (GMA, 98%), N,N,N',N'',N''-penta-methyldiethylenetriamine (PMDETA, 99%), ethanolamine (EA, 98%), anhydrous N,N-dimethylformamide (DMF, 99.8%), copper(I) bromide (CuBr, 99%), strepto-mycin, penicillin and 4',6-diamidino-2-phenylindole (DAPI) were obtained from Sigma-Aldrich Chemical Co. GMA was used after removal of the inhibitors. Type IV collagen targeted peptide (ColIV) KLWVLPKGGGC and rhodamine B-modified ColIV were obtained from ScilLight Biotechnology Co. (Beijing, China). Dulbecco's modified eagle medium (DMEM) was obtained from Hyclone Co. Trypsin, phosphate buffered saline (PBS) and fetal bovine serum (FBS) were purchased from Gibco Co. Trizol lysis buffer was obtained from Invitrogen Co. Diethyl pyrocarbonate (DEPC)-treated water was purchased from Solarbio Co. The synthetic microRNA-negative control (miR-neg, scramble miRNA molecules), scramble miRNA molecules labeled with the fluorescent dye Cy3 (miR-Cy3) and miR-145 were purchased from RIBBIO Co. (Guangzhou, China). Propidium iodide (PI) and Hoechst 33342 were purchased from Life Technologies Co. (Paisley, United Kingdom). Anti-KLF4 (ab75486 for in vitro WB, ab187962 for in vivo WB), anti-α-SMA (ab5694), anti-SM22α (ab14106), anti-myh11 (ab53219) were obtained from Abcam (Cambridge, MA). Anti-GAPDH (TA-08) was obtained from ZSJQB Co. (Beijing, China). Dye-conjugated secondary antibody was obtained from Rockland Immunochemicals Inc. (Gilbertsville, PA). The normal primers used in this work were obtained from Synbio Technologies Co. (Suzhou, China) and cDNA of miR-145 was obtained from ThermoFisher Scientific Co. (China).

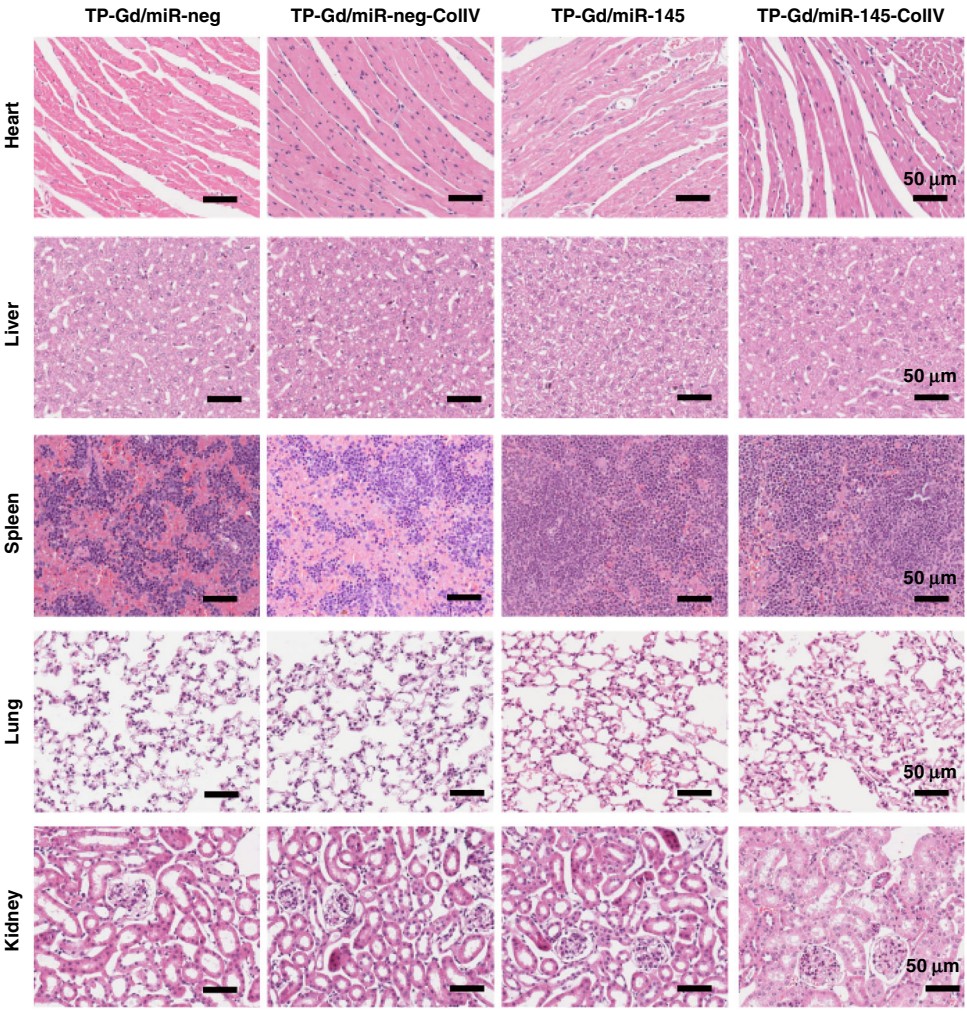

**Fig. 8** Representative photographs of H&E staining of paraffin-embedded sections of organs in TP-containing polycation/miRNA groups

**Synthesis of bromoisobutyryl-terminated TA (TA-Br)**. For the preparation of TA-Br, tannic acid (1.7 g, 1 mmol) and triethylamine (0.83 mL, 6 mmol) were dissolved in 20 mL of anhydrous DMF. The solution was stirred and cooled down to 0 °C in an ice bath. Then, BIBB (0.77 mL, 6 mmol) was slowly added dropwise into the reaction system. The reaction was allowed to proceed at room temperature for 24 h. The mixture was precipitated by excess of deionized water and then the pH of the system was adjusted to 8 by NaHCO$_3$. The precipitation was washed by purified deionized water for several times, and then freeze-dried to obtain the final powder, named as TA-Br.

**Synthesis of gadolinium chelated TA-Br (TA-Br-Gd)**. To ensure the success of ATRP, the initiator should be modified with gadolinium. TA-Br (1 g, 0.44 mmol) and GdCl$_3$·6H$_2$O (3.4 g, 9.1 mmol) were dissolved in 30 mL of DMF. The mixture was stirred at room temperature for 24 h. The product was precipitated by deionized water and then washed for several times. Finally, the product was freeze-dried to obtain the brown powder, named as TA-Br-Gd.

**Synthesis of TA-PGMA via ATRP**. TA-Br-Gd (200 mg, 0.07 mmol), GMA (2 mL, 7 mmol), and PMDETA (88 μL, 0.42 mmol) were prepared in a 50 mL flask containing 5 mL of DMSO. Before the addition of CuBr (60 mg, 0.42 mmol), the system should be degassed by nitrogen for 5 min to ensure the reaction under an oxygen-free environment. The polymerization was stirred at 30 °C for 30 min. Then the reaction was exposed to the air to terminate the polymerization. TA-PGMA was precipitated in an excess of methanol and purified by reprecipitation cycles with methanol to remove the extra reactants and by-product. The polymer was finally dried under reduced pressure, named as TA-PGMA ($M_n = 2.2 \times 10^4$ g mol$^{-1}$, PDI = 1.30). The detailed polymerization conditions are summarized in Supplementary Table 1.

**Synthesis of TA-PGEA via ring-opening reaction**. The preparation procedures of TA-PGEA as follow[55]. 0.2 g of TA-PGMA were dissolved in 5 mL of DMSO. The system was degassed by nitrogen for 5 min before adding 2 mL of EA. The reaction was stirred at 80 °C for 2 h. Crude product was purified using a dialysis membrane (MWCO 1000). The final product was obtained by freeze-drying method denoted as TA-PGEA (TP).

**Synthesis of gadolinium chelated TA-PGEA (TP-Gd)**. Due to the loss of cheated gadolinium during the preparation process of TP, the prepared TP was subsequently supplemented with more gadoliniums. 20 mg of TP was dissolved in 1 mL of deionized water. pH was adjusted to 6 by hydrochloric acid and then 15 mg of GdCl$_3$·6H$_2$O was added into system. After that, pH was adjusted to 8 by NaHCO$_3$. The reaction was stirred at room temperature for 24 h and the final product named as TA-PGEA-Gd (TP-Gd) was purified using a dialysis membrane (MWCO 1000) prior to lyophilization.

**Synthesis of TP-E, TP-ColIV, and TP-ColIV-RhB**. For the preparation of TA-PGEA/ED (TP-E), TA-PGMA (50 mg, 0.0023 mmol) was dissolved in 5 mL of DMSO. The system was degassed by nitrogen for 5 min before adding 15 μL of ED and 180 μL of EA. The reaction was stirred at 80 °C for 2 h. The resultant crude TP-E was purified using a dialysis membrane (MWCO 1000) prior to lyophilization.

For the preparation of TA-PGEA/ED-ColIV (TP-ColIV) and TA-PGEA/ED-ColIV-RhB (TP-ColIV-RhB), 10 mg of ColIV or 14 mg of rhodamine B-modified ColIV were dissolved in 1 mL of deionized water containing 1.7 mg of EDC·HCl and 1.0 mg of NHS, respectively. Each reaction was stirred at room temperature. 1 h later, 12 mg of TP-E was added into the reaction and then the system was stirred for another 24 h, The final products TP-ColIV and TP-ColIV-RhB were purified using a dialysis membrane (MWCO 10000) prior to lyophilization.

**Preparation of polycation/miRNA complexes**. All of the polycation solutions were prepared in water at a nitrogen concentration of 10 mM. Polycation-to-miRNA ratios were expressed as the molar ratios of nitrogen (N) in polycation to

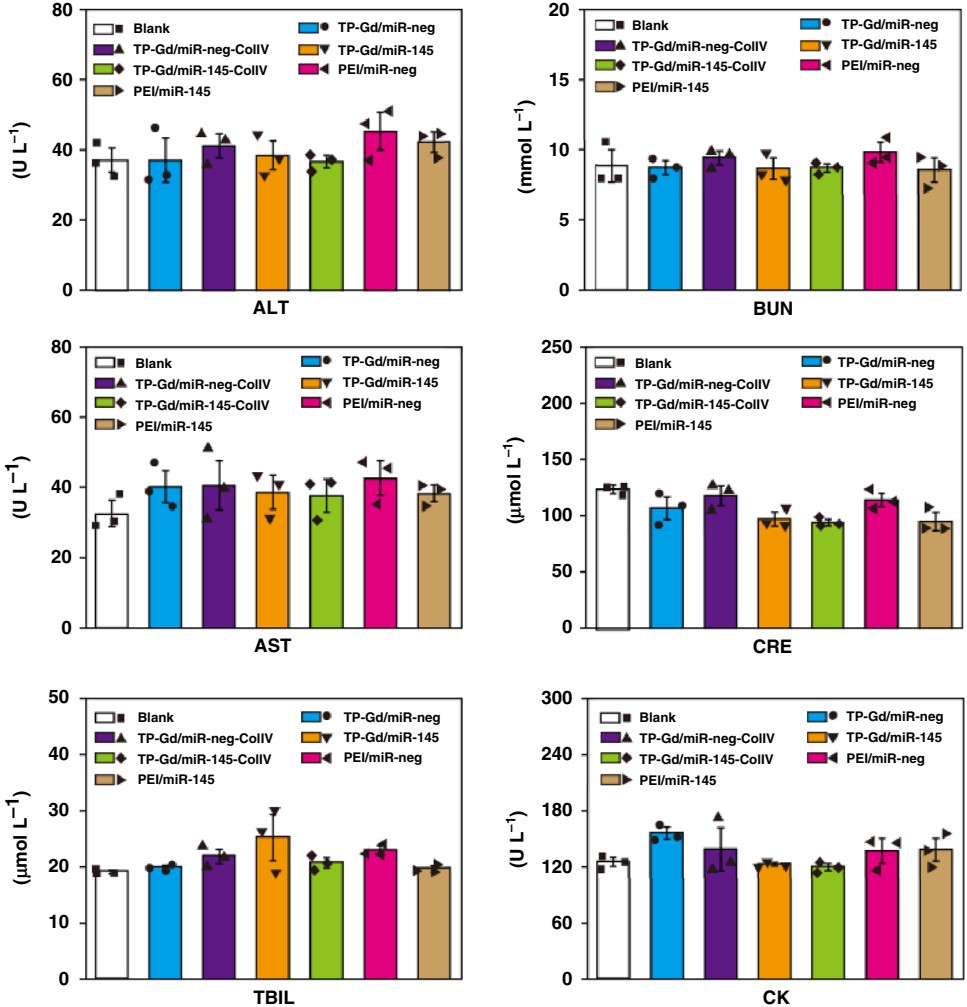

**Fig. 9** Plasma biochemical measurements of aspartate transaminase (AST), alanine transaminase (ALT), total bilirubin (TBIL), blood urea nitrogen (BUN), creatinine (CRE), and creatine kinase (CK). ($n = 3$ animals; error bars represent standard deviation; source data are provided as a Source Data file)

phosphate (P) in miRNA (or as N/P ratios). Most complexes with different N/P ratios were formed by mixing the polycation-containing solution and miRNA solution for 30 min before use except the targeting complex.

Electrostatic assembly was used for constructing the TP-Gd/miRNA-ColIV nanosystem. The TP-Gd solution (at a nitrogen concentration of 10 mM) and miRNA solution were firstly mixed at the N/P ratio of 5 for 15 min, and then the TP-ColIV solution (at a nitrogen concentration of 10 mM) was supplemented to be the final N/P ratio of 10 for the other 15 min.

**Physicochemical characterization**. [1]H NMR spectra were measured on a Bruker ARX 400 MHz spectrometer using DMSO-d6 (for TA-Br), CDCl₃ (for TA-PGMA) and D₂O (for TP, TP-E, and TP-ColIV) as the solvents with tetramethylsilane (Me₄Si) as an internal standard. Thermogravimetric analysis measurement of TA-Br-Gd was performed on a Thermal Gravimetric Analyzer (TG 209, NETZSCH). GPC measurement of TA-PGMA was performed on a Waters GPC system with DMSO as the eluent. The concentrations of Gd³⁺ in TP and TP-Gd were investigated using inductively coupled plasma mass spectrometry (ICP-MS, Thermo Scientific iCAP 6000 series). The fluorescence intensities of rhodamine B modified ColIV and TP-ColIV-RhB were characterized using a fluorescence spectro-photometer (Hitachi F-7000). Dynamic light scattering (DLS) measurements of polycation/miRNA complexes were performed with a Zetasizer Nano ZS equipped with a laser of wavelength 633 nm at a 173° scattering angle (Malvern Instruments, Southborough, MA). Atomic force microscopy (AFM) studies were carried out with the Dimension Icon model with a Nanoscope IIIa controller (Bruker, Santa Barbara, CA), where samples were imaged using the ScanAsyst mode. Gel electrophoresis was implemented in a Sub-Cell system (Bio-Rad Laboratories), and then a UV transilluminator and BioDco-It imaging system (UVP Inc.) was employed to record miRNA bands.

**Protein absorption assay**. 100 μL of TP-Gd solution (2.44 mg mL⁻¹ in DEPC-treated water) and 100 μL of PEI solution (0.43 mg mL⁻¹ in DEPC-treated water) were respectively mixed with 100 μL of miRNA solution (0.33 mg mL⁻¹ in DEPC-treated water). Then all the mixtures were kept for 30 min to form complexes. 40 μL of bovine serum albumin (BSA) solution (2 mg mL⁻¹ in DEPC-treated water) was added to the complex solution and shaken at 37 °C for 0.5–60 min. The supernatant was collected after high-speed centrifugation to remove the protein complexing with polycations. The BCA protein assay was used to determine the concentration of BSA in the supernatant.

**Hemolysis assay**. Hemolysis assay was performed on the blood of *C57BL/6J* mice (HFK Bioscience Co. Ltd., Beijing, China). Red blood cells (RBCs) were prepared previously. PEI, TP, TP-Gd, and TP-ColIV were added into 2% RBCs suspension with the concentration of 0.1 mg mL⁻¹ and 1 mg mL⁻¹, respectively. In addition, the RBCs incubated with deionized water and PBS were used as the corresponding positive and negative controls. All the suspensions were cultured at 37 °C for 4 h and then centrifuged at 1416 × g for 15 min. The absorbance of the supernatant was measured at 545 nm using a UV-vis spectrometer. All the hemolysis ratios were measured by the following equation.

$$\text{Hemolysis ratio}(\%) = \left[ \left( \text{OD}_{\text{test}} - \text{OD}_{\text{neg}} \right) \Big/ \left( \text{OD}_{\text{pos}} - \text{OD}_{\text{neg}} \right) \right] \times 100 \quad (1)$$

where OD$_{\text{test}}$, OD$_{\text{neg}}$, and OD$_{\text{pos}}$ represented the OD545 values of samples, negative control and positive control, respectively.

The morphological changes of RBSs treated with deionized water, PBS, TP, TP-Gd, and TP-ColIV at the concentration of 1 mg mL⁻¹ were observed. After incubation at 37 °C for 4 h, each RBCs solution was centrifuged at 14,160 × g for 20 min. Then the structures of collected RBCs pallet were captured by Confocal Laser Scanning Microscope (Leica, Germany), respectively.

**In vitro cytotoxicity assay**. The cell viability of different polycation/miRNA complexes at various N/P ratios was assessed using SMCs via Hoechst-PI assay. SMCs were extracted from the original generation and specific markers of this cells were tested from WB. SMCs were seeded in a 24-well plate at $5 \times 10^4$ cell well$^{-1}$ in 500 µL of normal medium (DMEM with 10% FBS, 100 units mL$^{-1}$ of penicillin and 100 µg mL$^{-1}$ of streptomycin). After 24 h incubation under normal conditions, the culture medium was replaced with fresh medium containing polycation/ miRNA complexes with 1 µg of miRNA (prepared in advance at the N/P ratios of 5–20) each well. 4 h later, the medium was substituted with fresh medium and cells were cultured for additional 20 h. After that, cells were washed by PBS and 200 µL of normal medium containing 2 µL of Hoechst 33342 (1 mg mL$^{-1}$) was added into each well. 15 min later, cells were washed by PBS and kept in 200 µL well$^{-1}$ of PI (50 mg mL$^{-1}$) at room temperature for 5 min. Finally, the fluorescence intensity was measured with the High Content Scanner (Molecular Devices, silicon Valley, CA) using a Mata Xpress software.

**In vitro cellular uptake assay**. miR-Cy3 was used for cellular uptake assay. $5 \times 10^4$ SMCs well$^{-1}$ were seeded in a 24-well plate with 500 µL well$^{-1}$ of normal medium. After 24 h incubation under normal conditions, the culture medium was replaced with fresh medium containing polycation/miR-Cy3 complexes with 1 µg of miR-Cy3 (prepared previously at the N/P ratios of 5–20) each well. 4 h later, the medium was substituted with fresh medium and cells were cultured for additional 20 h. The cultured cells were washed with PBS twice and DAPI (150 ng mL$^{-1}$ in PBS) was used to stain the nuclei of cultured cells for 10 min. Then, the cells were visualized and quantified with MD using a MataXpress software. The images of cellular uptake for some typical complexes were observed by Confocal Laser Scanning Microscope (CLSM, Leica, Germany).

**In vitro gene therapy with miR-145**. For the cellular internalization of polycation/ miR-145 complexes, the quantitative Real-Time PCR (qRT-PCR) and western blot (WB) measurements were carried out to determine the miRNA expression level and its target gene expression levels. For qRT-PCR measurement, 12-well plate ($1 \times 10^5$ SMCs well$^{-1}$) was prepared. The concentration of miR-145 was changed to 2 µg of the polycation/miRNA complexes per well. The procedures were as same as the procedures in in vivo experiments[56]. On the other hand, for WB measurement, 6-well plate ($2 \times 10^5$ SMCs well$^{-1}$) was prepared and the concentration of miR-145 was changed to 4 µg of the polycation/miRNA complexes per well. After transfection, the protein extracts were prepared with cell lysis buffer. Briefly, proteins were denatured by boiling (100 °C, 5 min), separated by SDS-PAGE, transferred to nitrocellulose membranes, and then incubated with the primary antibodies anti-KLF4 (1:1000) and anti-GAPDH (1:1000) at 4 °C overnight respectively and then with Dye-conjugated secondary antibodies (1:5000) for 1 h at room temperature. The images were quantified by the use of the Odyssey infrared imaging system (LI-COR Biosciences, Lincoln, NE).

**In vivo experiments**. All the wild-type *C57BL/6J* mice were received from Chinese Academy of Medical Sciences (Beijing, China). All the animal handling complied with animal welfare regulations of Capital Medical University. The Animal Subjects Committee of Capital Medical University approved the animal study protocol.

For in vivo cellular uptake assay, male wild-type (WT) *C57BL/6J* mice were used. All the three-week-old mice were fed with a regular diet and administered with BAPN in drinking water (1 g kg$^{-1}$ day$^{-1}$) for 2 weeks. Then TP-Gd/ miRNA-ColIV was injected via angular vein at a dose of 2.5 nmol miRNA at N/P ratio of 10 per mouse. By the detection of MRI, the mice under the same situation of vascular disorder were chosen for the in vivo cellular uptake assay. After 2 days, all the mice were treated with PEI/miR-Cy3, TP-Gd/miR-Cy3, and TP-Gd/miR-Cy3-ColIV at the N/P ratio of 10 via intravenous (i.v.) injection of angular vein. Each injection dose of the complex solution containing miR-Cy3 was 100 µL. In addition, mice were treated with miR-Cy3 in 100 µL of DEPC-treated water as the control. After 1, 12, and 24 h, all the mice were sacrificed. Thoracic aortas, hearts, livers, spleens, lungs, and kidneys were captured and analyzed by Xenogen IVIS spectrum (Caliper Life Science, America) with living image 2.11 software. Briefly, the aorta sections were firstly stained with DAPI. The stained sections were incubated with the anti-α-SMA (1:100) at 4 °C overnight for immunofluorescence staining and then with secondary antibodies at 37 °C for 2 h. The thoracic aorta sections were observed with CLSM to detect the entry of polycation/miR-Cy3 complexes into mouse thoracic aortas.

For the treatment of TAD, all the three-week-old mice were fed with a regular diet and administered with BAPN dissolved in drinking water (1 g kg$^{-1}$ day$^{-1}$) for 1 week. Then the mice were divided into six groups, which were treated with different miRNA complexes (PEI/miR-neg, PEI/miR-145, TP-Gd/miR-neg, TP-Gd/miR-145, TP-Gd/miR-neg-ColIV or TP-Gd/miR-145-ColIV) every three days until the end of fourth week. Three-week-old mice fed with saline were set as blank control. Each injection dose of complex solution was 100 µL, containing 2.5 nmol miRNA. The survival curves were recorded. Finally, mice were sacrificed for all the rest experiments. The typical images of thoracic aortic arches from different groups were captured by digital camera. Also, the incidence of TAD was counted according to the standard where the existence of intramural hematomas in thoracic aorta represents the formation of TAD. Part of the thoracic aortas were used for

H&E staining, elastin staining and KLF4-positive immunohistochemical staining, and the rest aortas were digested for qRT-PCR of miR-145, *α-SMA*, *SM22α*, and *myh11*. H&E staining, and elastin staining were performed[56]. Thoracic aorta arch sections underwent immunohistochemical staining with the anti-KLF4 (1:200) at 4 °C for 1 h and detected with 3,3′-diaminobenzidine for immunohistochemistry.

The procedure of in vivo WB analysis was as standard[56]. In this work, anti-KLF4, anti-SM22α (1:200) and anti-myh11 (1:200) were used as the primary antibody.

QRT-PCR analysis for miR-145 was described as follows in detail. The total RNAs of thoracic aortic arches were extracted by the Trizol reagen method (Invitrogen). Herein, the 2ddCt method was used to determine the delivery efficiency of miR-145[57,58]. To assess the miRNA-145 expression, the cDNA was synthesized using the TagMan miRNA Reverse Transcription kit (Applied Biosystems) for miRNA-145 and U6. Then, qRT-PCR was carried out with TaqMan gene expression (Applied Biosystem) following the manufacturer's protocols. MiR-145 expression was normalized to U6. To detect mRNA expression of KLF4 associated genes, the aliquots (2 µg of total RNA) were used for first-strand cDNA synthesis with moloney murine leukemia virus reverse transcriptase (Promega, Southampton, UK). Aliquots (2 µL of reaction mixture) were amplified with 10 µL of SYBR Green PCR Master Mix and 1 µmol L$^{-1}$ primers. Supplementary Table 2 showed the primers used. Amplification was at 95 °C for 5 min, 95 °C for 45 s, and 60 °C for 1 min for each step for 45 cycles. The comparative cycle threshold method was used for the relative quantification of gene expression as standard. As shown in Supplementary Table 3, all the average Ct values of miR-145 in different groups ranged from 19 to 24 which demonstrated the high reliability of this method. Other QRT-PCR analysis for mRNA were performed as standard[56].

**In vitro and in vivo MRI**. MRI experiments were performed using a 7.0T MRI instrument (Biospec 70/20USR7.0T Bruker). To investigate the MRI abilities of TP and TP-Gd, each of solutions at different concentrations ($Gd^{3+} = 0$, 0.05, 0.1, 0.15, 0.2, 0.25 mM) were detected. All the solutions in 0.2 mL tubes were placed in a rat head/mouse body coil (Agilent Technologies) using a $T_1$ mapping RARE sequence (repetition time = 400, 800, 1500, 2500, 4000 ms; echo time = 7 ms; field of view = 4.0 cm$^2$; matrix = 256 × 256; number of excitation = 2.0; slice thickness = 1 mm; slice gap = 0 mm). The $R_1$ value was measured using a Matlab software.

For the intracellular MRI, approximately $5 \times 10^6$ SMCs were separately seeded and incubated in 25 mL cell culture flasks for 24 h. Then, the medium was replaced with 5 mL of fresh medium containing TP-Gd/miRNA, TP-Gd/miRNA-ColIV or MultiHance ($Gd^{3+} = 0$, 5, 10, 15, and 20 µM), respectively. After 4 h incubation at 37 °C, the cells were washed by PBS for 3 times, lysed by pancreatin, and then precipitated at the tube bottom after centrifugation. The MR imaging experiments were performed on the same 7.0 T MRI instrument. The $T_1$ mapping RARE sequence was same as above. The $T_1$ relativities were calculated using a linear fitting of the inverse $T_1$ relaxation time as a function of $Gd^{3+}$ concentrations.

For the preparation of in vivo MRI experiments, the metabolic efficiency about different complexes was measured by ICP-MS. *C57BL/6J* mice were divided into three groups (treated with saline, TP-Gd/miR-neg, and TP-Gd/miR-neg-ColIV, respectively). All the urine of the mice was collected at the same time in the two days.

In vivo MRI experiments were divided into two parts: early diagnosis and monitoring. For the diagnostic assay, BAPN models of two weeks were prepared for MRI detection. The examinations were performed by the same MRI instrument with a mouse head/mouse body coil. All the mice were anesthetized with a 2% isoflurane-oxygen mixture in an isoflurane induction chamber. A respiratory sensor connected to a monitoring system (SA Instrument, Stony Brook, NY, USA) was placed on the abdomen to monitor the respiration rate and depth. Images were acquired using a $T_1$-weighted black blood sequence (repetition time 1/4 = 923.06 ms; echo time 1/4 = 2.74 ms; inversion time $T_1$ = 773.06 ms; number of excitations = 2.0; average 1/4 = 3; flip angle = 30°; field of view = 3.0 cm$^2$; matrix = 256 × 256; scan duration with respiratory gating = 20 min). After MRI plain scan, TP-Gd/ miR-neg was injected via the angular vein at a dose of 20 µg of $Gd^{3+}$ per mouse. 0.5 h later, the images were acquired. 2 days later, all these mice primarily underwent plain scanning to confirm no residual MRI signals. Then they were injected with TP-Gd/miR-neg-ColIV at the same dose and the images were acquired after 0.5 h accumulation.

For the monitoring assay, the mice were fed by BAPN for one week. Then, these mice were separately injected with TP-Gd/miR-neg-ColIV (as the control group) and TP-Gd/miR-145-ColIV (as the experimental group) every three days until the end of the fourth week. In the third week and fourth week, the mice were detected by MRI to evaluate the severity of TAD using TP-Gd/miR-neg-ColIV (at a dose of 20 µg of $Gd^{3+}$ per mouse) as the detection agent. The MRI detection measurements and $T_1$-weighted sequence were the same as the above diagnostic assay.

All the MR images were analyzed using OsiriX DICOM Viewer (Geneva, Switzerland), using the spinal cord as a landmark. After each injection, ROIs were drawn around the enhancement area within the TAD wall. Average signal intensity was measured over three slices throughout the dissection. For pre- and post- injection, SNR (signal to noise ratio) of wall and muscle can be acquired (SNR$_{wall}$ and SNR$_{muscle}$). Finally, contrast to noise ratio (CNR) and normalized

percentage signal enhancement (%NSE) were calculated according to the following equations.

$$CNR_{WM} = SNR_{wall}/SNR_{muscle} \qquad (2)$$

$$CNR_{WM-pre} = \frac{SNR_{wall-pre}}{SNR_{muscle-pre}} \qquad (3)$$

$$CNR_{norm} = \frac{CNR_{WM}}{CNR_{WM-pre}} \qquad (4)$$

$$\%NSE = \left[\frac{(CNR_{WM} - CNR_{norm})}{CNR_{norm}}\right] \times 100 \qquad (5)$$

**In vivo cytotoxicity assay**. To determine the possible organ toxicity, the tissues including heart, liver, spleen, lung, and kidney of different groups were embedded for H&E staining after the in vivo treatment. Also, 48 h after the final injection of different complexes, mice blood in different groups (which were starved for 12 h) was collected through cardiac puncture. The plasma samples were assayed for the tests of ALT, BUN, AST, CRE, TBIL, and CK using an autoanalyser (RA 1000; Technicon Instruments, NY, USA). CK activities in plasma samples were measured by commercially available kits. Blinded histopathology scoring assay was performed by three pathologists from Anzhen hospital (Beijing, China) according to a 12-point histologic grading system[59,60].

**Statistical analysis**. All experiments were repeated at least three times. A normality test (Kolmogorov–Smirnov test) was primarily used for measurement data. Normally distributed data were analyzed using parametric testing. Unpaired Student's $t$-tests was for two groups and one-way analysis of variance (ANOVA) was conducted followed by Bonferroni's post hoc test for more than two groups. Non-normally distributed data were analyzed using non-parametric testing: Mann-Whitney $U$ test for two groups and Krushal–Wallis $H$ test for more than two groups. If the measurement data followed a normal distribution, the data are expressed as means ± standard deviation (SD); if not, the data are expressed as median values ± interquartile range (IQR). In all tests, statistical significance was set at $P < 0.05$. In all the pictures, $P < 0.05$ represented one star, $P < 0.01$ represented two stars, $P < 0.001$ represented three stars and $P < 0.0001$ represented four stars.

## Data availability
All relevant data are available upon reasonable request. The source data underlying Figs. 2a, b, 3a, b, e–f, 4a, 5b–d, 6a, c, 7b–d, 9 and Supplementary Tables 1, 3 and Figs. 1, 3a–c, 4–8, and 11–14 are provided as a Source Data file.

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

## Acknowledgements
This work was supported in part by the National Natural Science Foundation of China (Grant Numbers 51733001, 51829301, 91339000, and 91539121) and the Fundamental Research Funds for the Central Universities (Grant No. XK1802-2). Animal experimental protocols were approved by the Animal Care and Use Committee of Capital Medical University.

## Author contributions
C.X., Z.Z.Z., Y.L.L., J.D. and F-J.X conceived the project and planned experiments, C.X. synthesized materials and conducted their structural analysis. B.R.Y., J.J.N. and G.C. assisted with the characterizations of materials. Z.Z.Z. and K.X. assisted with the in vivo experiments. S.J.L. assisted with MRI experiments. C.X., Z.Z.Z., Y.L.L., J.D. and F-J.X co-wrote the paper. All authors discussed the results during manuscript preparation.

## Additional information

**Competing interests:** The authors declare no competing interests.

