## [Peer Review File · Nature Communications]

Reviewers' Comments:

Reviewer #1:

Remarks to the Author:

Xu et. al present a cationic nanoformulation as a targeted theranostic to co-deliver mir-145 microRNA (therapeutic) and gadolinium (diagnostic MRI contrast agent) for the treatment and monitoring of thoracic aortic dissection (TAD), respectively. The nanoformulation was synthesized from a gadolinium chelated and bromoisobutyryl modified tannic acid initiator used in the atom transfer radical polymerization of 4-arm poly(glycidyl methacrylate). A portion of this 4-arm TA-PGMA was then modified with ethanolamine in a ring-opening reaction and subsequently chelated with additional gadolinium to produce TA-PGEA-Gd. The remaining 4-arm TA-PGMA was reacted with ethanolamine and ethyl diamine to produce TP-PGEA/ED. The primary amines introduced by ED were then utilized in an amidation reaction to conjugate a Type IV collagen targeting peptide yielding TP-CollIV. TA-PGEA-Gd was complexed with mir-145 to produce nanocomplexes that were then surface modified with the TP-CollIV to yield the targeted nanomissile formulation TP-Gd/mir-145-L. The authors demonstrated the ability of these nanomissiles to effectively complex mir-145 and analyzed the charge ration (i.e., N/P) dependent effects on nanomissile size and surface charge. Protein adsorption, hemolysis, and in vitro cytotoxicity demonstrated enhanced stability, hemocompatibility, and biocompatibility of the nanomissile formulation compared to complexes formulated with the standard benchmark of polyethyleneimine (PEI). Targeted nanomissiles were found to enhance cellular uptake leading to increased mir-145 expression and concomitant decreased KLF4 expression in vitro. An in vivo cellular uptake assay was performed in a BAPN mouse model of TAD demonstrating that targeted nanomissiles significantly enhanced mir-145 delivery to the thoracic aorta. Furthermore, the targeted nanomissiles were found to improve survival, decrease TAD incidence, and prevent thoracic aorta dilation in vivo in the mouse model of TAD. The authors demonstrated that the targeted nanomissiles enhanced mir-145 expression in vivo which corresponded to enhanced expression of the contractile smooth muscle phenotypic markers alpha-SMA, SMA22-alpha, and MYH11 and decreased expression of KLF4. The authors also demonstrated the ability of these nanomissiles to serve as a targeted contrast agent for the early diagnosis and monitoring of TAD pathogenesis in the in vivo mouse model of TAD. Finally, the authors demonstrated that these nanomissiles are non-toxic based on immunohistochemical analysis of organ tissues and blood analysis of organ-associated toxicity.

This work is novel and well organized. However, the thoroughness and robustness (appropriate controls and level of quantitation) of some data sets is lacking, and portions of the text are unclear/poorly worded. We recommend that the authors address the following critiques:

Major Critiques:

1. Throughout the results there are a number of instances in which authors make vague and general statements about the results without directly stating quantifiable metrics. For example on page 7 lines 172-173 the authors state that, "Tp-Gd/miRNA exhibited better serum-tolerant ability and demonstrated significantly lower protein adsorption" without stating what the difference actually was / providing a quantifiable metric for these readouts. There are a number of other places where this applies as well: "hemolysis ratios were much lower" on lines 177-178; "expression amounts of KLF4 were decreased" on lines 226-227; etc. Clearly and directly stating quantifiable metrics in the text will improve the clarity and robustness of the work.
2. The authors do not discuss the importance/relevance of tannic acid in the nanomissile formulation. There is an abundance of very recent high profile publications highlighting the ability of tannic acid to target drugs to cardiac tissue. Discussing the reasoning and function of tannic acid in this regard would further strengthen the introduction to this work.
3. The paper refers to this as a layer by layer assembly. This typically refers to multiple layers of alternating charged polymers or components with some of type of interaction. This is more accurately a simple two step fabrication procedure. I'm not sure that LBL scientists would consider this accurate.
4. The paper should include some discussion of the timing of such a therapy if applied in people. It

may also be worth discussing cross-species feasibility. The model used is aggressive and fast-acting. How well does this apply to the gradual, chronic development of TAD in natural circumstances?

5. The authors state that "Mir-145 would stabilize vascular structure and prevent deterioration of TAD" in lines 94-95 of the introduction without discussing the underlying mechanisms / function of mir-145 and how it relates to TAD pathogenesis. A more detailed description of the disease and justification for the use of mir-145 in the context of the disease would further strengthen this work.

6. The specificity of the collagen IV targeting peptide is not proven. The particles have quite high zeta potential, which is typically bad for targetability in vitro (because the positive charge carries the particles into everything) and also for in vivo circulation time / targetability. The authors should show immunocytochemistry to prove that collagen IV is on the SMC surface and better explain how targeting a secreted protein may work. They should also prove that the targeting does not work in a control cell type that does not express collagen IV.

7. The authors present data on delivery to the thoracic aorta and clearance in the urine but do not present any whole animal biodistribution data. Where else do the nanomissiles accumulate and are there any potential off-target effects that could arise from systemic administration of this formulation? Whole animal biodistribution data (i.e., relative amounts delivered to primary organs) and vascular pharmacokinetics (circulation time) should be provided in this work.

8. PEI complexes are utilized as a standard control for many of the experiments but not for the data shown in figure 1e-f or the in vivo efficacy data in figures 4 and 5. Why were these controls omitted from these studies? Having consistent controls across all data presented would further strengthen this work.

9. The duration of BAPN treatment to induce TAD in the various in vivo models is inconsistent (i.e., 2 weeks for in vivo uptake, 4 weeks for gene therapy studies, one week for blood toxicity assessments) – why? Having consistent conditions across the data sets shown would further improve the strength of this work.

10. On lines 269-270, the authors state that "no dissection was obviously occurred in the Tp-Gd/mir-145 group" whereas the data shown in figure 4d shows an incidence of TAD around 20% for this treatment group. The text should be modified to accurately represent the results.

11. The efficacy data show in Figure 4 does not include a healthy, non-TAD control group. Including this control would further strengthen the data, specifically the thoracic aorta diameter data shown in 4e. Similarly, the IHC analysis of organ toxicity in figure 7 does not include a healthy, non-TAD control group for comparison.

12. Protein quantification by western on multiple treated animals should be provided for KLF-4 and the protein markers related to contractile vs synthetic SMC phenotype to improve the Figure 5 data set.

13. Blinded histopathology scoring should be done on multiple animals as opposed to showing a single H&E image per organ.

Minor Critiques:

1. The use of the term "nanomissile" is not really clear and is unnecessary.

2. There is a lack of discussion and justification for the animal model of TAD utilized. How does BAPN induce TAD and how closely does it match TAD in humans?

3. The authors mention in the introduction that, "TAD is one of the most devastating diseases" but do not provide any quantifiable metrics in terms of prevalence/incidence or financial burden. Detailing the impact of this disease would further strengthen the significance of this work.

4. There are a number of typographical and grammatical errors throughout the text. For example:

a. Typos: "benefiting" line 71, page 3; "in details" line 102, page 4; "condensed" line 147 page 6; "MRI plain scanning" line 325, page 13; etc.

b. Grammar / diction errors: The word "charged" should be inserted between "negatively" and "nucleic" in line 75; remove the word "to" prior to "2" on line 147; the word "living" should be changed to "live" on line 234; "fluorescent signal of PEI/Cy3 did not be detected obviously" should be change to "could not be detected"; "and no fluorescent signal in thoracic aortic arch" lines 236-237 is an incomplete sentence; "were showed" in line 267 should be changed to "are shown"; etc.

Overall, the text should be thoroughly proof read and the grammatical errors corrected.

5. Articulate and support this statement: "Powerful..." line 74, page 3.

6. The dose and route of administration should be stated in the paper body, rather than only in the supplement. Dose is typically presented in mg/kg of the RNA. The quantity is low enough to argue for the value of such a therapy, in spite of the frequent dosing required.

7. The authors state that the "rich hydroxyls can effectively balance the surface potential of complexes and reduce the deleterious effects of excess positive charges" lines 186-187. However, the results shown in Figure 1b show that there are no significant differences in surface charge between the nanomissiles and PEI complexes. This statement is not justified by the data shown and should be removed or altered to accurately reflect the data.

8. Several particle formulations are referred to with -L (like TP-Gd/miRNA-L), which is never identified explicitly. I assume this means it has undergone layer-by-layer self-assembly? The discussion in lines 153-157 comes a little late given the earlier references, and

9. The bars in Figure 1b surely do not represent the PDI. Should probably at least include the PDI for these four in Table S1.

10. The authors state that the "fluorescent signal of PEI/miR-Cy3 did not be detected obviously" on line 238, however figure 3b does show apparent signal in the PEI/miR-Cy3 group, albeit significantly less than the nanomissile treatment groups. The text should be modified to accurately address this.

11. The quantification of delivery in figure 3a is unclear – the y axis and texts mention the relative ROI but do not detail what is actually being measured. Is it the average fluorescence intensity of the ROI that is being compared? Is there any normalization of the fluorescent signal to the area of the ROI? These issues should be clarified in the text.

Reviewer #2:

Remarks to the Author:

In this article, Xu et al. have designed nanoparticle TP-Gd and demonstrated its potential in the delivery of microRNA mimetics to the site of thoracic aortic dissection (TAD). Briefly, they clearly introduced the methodology to synthesize this nanoparticle. Notably, the ATRP and layer-by-layer self-assembly enhance the stability and targeting efficacy of this TP-Gd. In vitro cellular uptake assay shows that SMC uptakes miR-145 conjugated TP-Gd and the delivered miR-145 mimetics successfully suppress target KLF4, which suggests that miR-145 can be released from TP-Gd particles and loaded on RISC complex. The MRI result suggested that TP-Gd is a good tool to non-invasively detect TAD site. Overall, the data is nicely presented and manuscript is well-written. However, more data are required to confirm the most valuable part of this study—if TP-Gd deliver microRNA "directly" to smooth muscle cells at the site of TAD in animals. Below are my comments:

1. In figure 3b, authors should provide higher magnified pictures to demonstrate that TP-Gd/miR-Cy3-L is indeed "in" the cytoplasm of smooth muscle cells. Current level of magnification can only demonstrate that the TP-Gd/miR-Cy3-L was delivered to TAD region.

2. A followed up question to the first point, what is the organ distribution of TP-Gd/miR-Cy3-L? Author should image whole mouse body to evaluate if TP-Gd/miR-Cy3-L are specifically delivered to TAD, instead of other organs.

3. In figure 5a, the amount of miR-145 being delivered to SMC at TAD site should be presented as absolute amount (i.e. copy number), instead of relative amount determined by 2ddCt method; otherwise, it is difficult to evaluate the delivery efficacy.

4. In figure 5c, where exactly on the thoracic arch the KLF4 expression was reduced? Authors should take higher magnified images to demonstrate the IHC result and use arrow or dashed line to label or highlight the site showing high KLF4 in control groups but less KLF4 in miR145 groups.

5. In figure 5c, the overall staining of KLF4 looks pretty weak in control group, authors should determine the protein level of KLF4 with western blot to evaluate the effect of miR-145 delivery.

6. In figure 6b, the labeling of nanoparticle used in control groups should be consistent with figure 6d (i.e., TP-Gd/miRNA should be labeled as TP-Gd/miRNA-neg). It is confusing to see different name for control groups.

7. In figure 6d, the signal in the background area (not included in dash lined area) seems to be stronger in negative control group than that in miR-145-L group. To justify the imaging quality, authors should provide lower magnified images for all of the TAD images.

8. Liver function test should be performed at 24-48 hours after administration of nanoparticle. One hour is too short to detect changes of these common liver and kidney function biomarkers.

9. In figure 7, author should also use lower magnified images for heart, liver and kidney to fairly evaluate the damage or inflammation through out the organ, if there is any.

Reviewer #3:

Remarks to the Author:

The English usage needs some attention.

The molecular biology was difficult to follow for this clinical reviewer. I am certain that experts in this area will comment on those aspects.

From the clinical point of view, I have the following observations:

1) You seem to have the misconception that TAD (thoracic aortic dissection) is a chronic event. Certainly, the aortic wall does deteriorate over time. However, TAD is an INSTANTANEOUS event that occurs in a vulnerable aorta. Please review your entire manuscript, especially the clinical observations, with this in mind.

2) You seem to have the misconception that it is difficult to diagnose TAD. (You say: "diagnostic methods cannot clearly demonstrate the location and size of TAD.[50-55]". That is completely untrue. Cardiac echocardiography, computed tomography, and magnetic resonance imaging are all capable of detecting TAD with a very high level of accuracy.

3) You seem to have the misconception that TAD is clinically silent (You say: "TAD develops without obvious clinical manifestation, deteriorates quickly and finally lead to dissection rupture and organ ischemia.) TAD has very specific and dramatic clinical manifestations (chest or back pain, organ ischemia).

4) Your references do not always seem pertinent where you have cited them. Please check every one.

Congratulations on your very promising findings, which may hold great clinical importance.

Thank you very much for your e-mail about our manuscript entitled “**Multifunctional cationic nanosystems for precise nucleic acid therapy of thoracic aortic dissection**” (Manuscript ID: NCOMMS-18-15152332-T).

We are very grateful to Reviewers for their kind comments and suggestions on our work. As you can see, this manuscript has been revised substantially. More data were provided to address the reviewers concerns. The revised parts have been highlighted in yellow in the manuscript. Detailed response to the referees’ comments and the changes made to the manuscripts are listed as follows:

REVIEWER REPORT(S):

Referee: 1

Xu et. al present a cationic nanoformulation as a targeted theranostic to co-deliver mir-145 microRNA (therapeutic) and gadolinium (diagnostic MRI contrast agent) for the treatment and monitoring of thoracic aortic dissection (TAD), respectively....This work is novel and well organized. However, the thoroughness and robustness (appropriate controls and level of quantitation) of some data sets is lacking, and portions of the text are unclear/poorly worded. We recommend that the authors address the following critiques:

Major Critiques:

1. Throughout the results there are a number of instances in which authors make vague and general statements about the results without directly stating quantifiable metrics. For example on page 7 lines 172-173 the authors state that, “Tp-Gd/miRNA exhibited better serum-tolerant ability and demonstrated significantly lower protein adsorption” without stating what the difference actually was / providing a quantifiable metric for these readouts. There are a number of other places where this applies as well: “hemolysis ratios were much lower” on lines 177-178; “expression amounts of KLF4 were decreased” on lines 226-227; etc. Clearly and directly stating quantifiable metrics in the text will improve the clarity and robustness of the work.

Response: Thanks for your kind advice. As suggested, in this revision we added some clear and direct statements of quantifiable metrics about the important results (see **Lines 20-21, 25 Page 8; Line 1, Page 9; Lines 21-22, Page 10; Line 10, Page 13 and Lines 10-11, Page 16;** etc).

2. The authors do not discuss the importance/relevance of tannic acid in the nanomissile formulation. There is an abundance of very recent high profile publications highlighting the ability of tannic acid to target drugs to cardiac tissue. Discussing the reasoning and function of tannic acid in this regard would further strengthen the introduction to this work.

Response: Clarified accordingly. Tannic acid is a natural molecule which protect hear and vessels from cardiovascular disease. It was reported the TA modified collagen could specifically target to heart (**Nat. Biomed. Eng. 2, 304-317 (2018), Reference**

41). Herein, we used TA because of its excellent properties of complexing ability and biocompatibility. More information about TA was added in this revision (see **Lines 21-24, Page 4 and References 41-43**)

3. The paper refers to this as a layer by layer assembly. This typically refers to multiple layers of alternating charged polymers or components with some of type of interaction. This is more accurately a simple two step fabrication procedure. I'm not sure that LBL scientists would consider this accurate.

Response: Thanks for your comments. We discussed with LBL scientists. They also think here the term 'layer by layer assembly' is inaccurate. Thus, in this revision, 'electrostatic assembly' was used to replace 'layer by layer assembly'. (see **Line 9, Page 2; Line 19, Page 4; Line 24, Page 7 and Line 18, Page 5**)

4. The paper should include some discussion of the timing of such a therapy if applied in people. It may also be worth discussing cross-species feasibility. The model used is aggressive and fast-acting. How well does this apply to the gradual, chronic development of TAD in natural circumstances?

Response: Thanks for your kind advice. Normally, aortic dissection develops rapidly and expands quickly which would lead to a rupture and organ ischemia. It is difficult to detect and treat pre-TAD via traditional medical technology expect for taking a large number of diagnostic imaging tests. Thus, this therapy is suitable for the people which are susceptible to TAD, such as marfan patients. These patients could be detected by this treatment before TAD. When high MRI signal appears, they could be intervened by this therapy. The corresponding information was added in this revision (see **Lines 10-11, Page 18**).

5. The authors state that "Mir-145 would stabilize vascular structure and prevent deterioration of TAD" in lines 94-95 of the introduction without discussing the underlying mechanisms / function of mir-145 and how it relates to TAD pathogenesis. A more detailed description of the disease and justification for the use of mir-145 in the context of the disease would further strengthen this work.

Response: As suggested, more information about mir-145 was added in this revision. MiR-145, the upstream important factor for the regulation of KLF4, is very promising for the treatment of TAD.^[16-18] Among vascular diseases, KLF4 is an essential transcription factor related with the phenotypic switching of smooth muscle cells (SMCs). With the promotion of miR-145, KLF4 level will be decreased and SMCs would maintain contractile phenotypes, finally benefitting the protection from TAD.^[19,20] (see **Lines 15-20, Page 3 and References 16-20**) As the important upstream factor of KLF4, miR-145 is the suitable highly conserved small noncoding RNA molecule for the treatment of TAD.

6. The specificity of the collagen IV targeting peptide is not proven. The particles have quite high zeta potential, which is typically bad for target ability in vitro (because the positive charge carries the particles into everting) and also for in vivo

circulation time/target ability. The authors should show immunocytochemistry to prove that collagen IV is on the SMC surface and better explain how targeting a secreted protein may work. They should also prove that the targeting does not work in a control cell type that does not express collagen IV.

Response: Clarified accordingly. The specificity of collagen IV targeting peptide (CollIV) was proven using the refereed immunocytochemistry in our earlier work (**Theranostics 8, 437-449 (2018), Reference 35**). CollIV can effectively target to SMCs which express collagens IV as extracellular matrix. In this revision, more information about the specificity of CollIV was added (see **Lines 9-12, Page 4 and Reference 35**).

7. The authors present data on delivery to the thoracic aorta and clearance in the urine but do not present any whole animal biodistribution data. Where else do the nanomissiles accumulate and are there any potential off-target effects that could arise from systemic administration of this formulation? Whole animal biodistribution data (i.e., relative amounts delivered to primary organs) and vascular pharmacokinetics (circulation time) should be provided in this work.

Response: As suggested, we performed the experiment of the whole animal biodistribution at different time points of 0-24 h. The corresponding data were added in this revision (see **Figure S8 in Supporting Information**). Under the same dose of miR-Cy3, the fluorescent signal of thoracic aorta in the TP-Gd/miR-145-CollIV group due to its targeting effect was the strongest among all the groups at each time point. With increasing time, the fluorescent signals of miR-Cy3 increased in livers and kidneys, demonstrating the metabolic process. (see **Lines 21-25, Page 11 and Lines 1-2, Page 12**)

8. PEI complexes are utilized as a standard control for many of the experiments but not for the data shown in figure 1e-f or the in vivo efficacy data in figures 4 and 5. Why were these controls omitted from these studies? Having consistent controls across all data presented would further strengthen this work.

Response: Thanks for your suggestion. As suggested, in this revision we supplemented all the experimental data of PEI *in vitro* and *in vivo*. For *in vitro* experiments, PEI transfected a little amount of miR-145 into SMCs, and no obvious change in the expression levels of KLF4 was observed. (see **Figure 2e and Figure S7 in Supporting Information**; see **Lines 9-12, 20-24, Page 11**). For *in vivo* experimental data, there is no difference between the PEI/miR-neg and PEI/miR-145 groups. The mice in these PEI groups exhibited low survival ratios and high incidence of TAD (see **Figure 4b and 4d**; see **Lines 21-24, Page 12 and Lines 8-10, Page 13**). H&E and elastin stainings also illustrated no therapeutic effect in the PEI-involving groups (see **Figure S9b in Supporting Information**; see **Lines 19-22, Page 13**). Based on the data of IHC analysis (see **Figures S10, S11 in Supporting Information**; see **Lines 7-13, Page 14**), qRT-PCR (see **Figure 5a**; **Lines 5-7, Page 14**) and WB (see **Figure 5c and Figure S12 in Supporting Information**; see **Lines 15-18, Page 14**) for different markers, it is found that PEI could not effectively transfect miR-145 into

thoracic aortas and decreases the expression levels of KLF4. In addition, the corresponding data of PEI about long-term toxicity to organs were also added (see **Figure 8 and Figure S16 in Supporting Information**; see **Lines 9-10, 16-17, Page 17**).

9. The duration of BAPN treatment to induce TAD in the various in vivo models is inconsistent (i.e., 2 weeks for in vivo uptake, 4 weeks for gene therapy studies, one week for blood toxicity assessments) – why? Having consistent conditions across the data sets shown would further improve the strength of this work.

Response: Clarified accordingly. The targeting ability of nanosystems was measured during the dissection formation. Thus, the two-week TAD model with no obvious dissection is suitable for this *in vivo* uptake study (see **Lines 5-7, Page 11**). Four-week BAPN model includes the process from pre-TAD to TAD, which is suitable for evaluating the therapeutic effect (see **Lines 13-14, Page 12**). For blood toxicity assessments, as suggested, four-week BAPN model was used to replace one-week model. The updated data about blood toxicity assessments were added in this revision. (see **Figure 8, Lines 16-17, Page 17 and Lines 22-23, Page S13; Line 1, Page S14 in Supporting Information**).

10. On lines 269-270, the authors state that “no dissection was obviously occurred in the Tp-Gd/mir-145 group” whereas the data shown in figure 4d shows an incidence of TAD around 20% for this treatment group. The text should be modified to accurately represent the results.

Response: Modified accordingly. (see **Lines 6-7, Page 13**).

11. The efficacy data show in Figure 4 does not include a healthy, non-TAD control group. Including this control would further strengthen the data, specifically the thoracic aorta diameter data shown in 4e. Similarly, the IHC analysis of organ toxicity in figure 7 does not include a healthy, non-TAD control group for comparison.

Response: Thanks for your suggestion. As suggested, in this revision we supplemented the corresponding experimental data of healthy, non-TAD mice. The mice in the healthy group exhibited 100% survival ratio, no incidence of TAD was observed and the diameter data of their thoracic aortas can be used for evaluating the therapeutic effect. (see **Figure 4b, d, e and Figure S9a in Supporting Information**; see **Lines 20-21, Page 12 and Lines 3-4, 11-12, Page 13**). H&E and elastin stainings of the healthy, non-TAD control group also provide the negative control for the other groups (see **Figure S9b in Supporting Information**; see **Lines 19-22, Page 13**). The data of IHC (see **Figures S10, S11 in Supporting Information; Lines 7-13, Page 14**) and WB (see **Figure 5c and Figure S12 in Supporting Information**; see **Lines 15-18, Page 14**) of the healthy, non-TAD control group were also provided. In addition, the H&E staining and data of plasma biochemical measurements of the healthy, non-TAD control group were also added in this revision for the comparison with other groups (see **Figure 8 and Figure S16 in Supporting Information**; see **Lines 9-10, 16-17, Page 17**).

12. Protein quantification by western on multiple treated animals should be provided for KLF-4 and the protein markers related to contractile vs synthetic SMC phenotype to improve the Figure 5 data set.

Response: As suggested, in this revision we supplemented the protein quantification by WB for KLF-4 and protein markers (related to contractile SMC phenotype) (see **Figure 5c, Figure S12 in Supporting Information**; see **Lines 15-18, 24-25, Page 14** and **Lines 1-2, Page 15**). The results of WB were consistent with those of IHC analysis and qRT-PCR.

13. Blinded histopathology scoring should be done on multiple animals as opposed to showing a single H&E image per organ.

Response: Thanks for your kind advice. As suggested, blinded histopathology scoring assay was performed by three pathologists from Anzhen hospital (Beijing, China) according to a 12-point histologic grading systems (see **References S6, S7 (Proc. Natl. Acad. Sci. 91, 947-951 (1994) and toxins. 8, 259 (2016))** in **Supporting Information; Lines 17-19, Page 17**).

Minor Critiques:

1. The use of the term “nanomissile” is not really clear and is unnecessary.

Response: Thanks for your suggestion. We have changed “nanomissile” to “nanosystem” throughout the manuscript.

2. There is a lack of discussion and justification for the animal model of TAD utilized. How does BAPN induce TAD and how closely does it match TAD in humans?

Response: Added accordingly. BAPN is an inhibitor of lysyl oxidase which would lead to the typical degradation of extracellular matrix (ECM) and the loss of SMCs, finally resulting in TAD. Also, the symptom and effect of BAPN-induced TAD were consistent with the observations in humans based on earlier works (see **References 48,49 (Vascular 21, 287-292 (2013) and Circulation 126, 3070-3080 (2012))**; see **Lines 3-5, Page 11**).

3. The authors mention in the introduction that, “TAD is one of the most devastating diseases” but do not provide any quantifiable metrics in terms of prevalence/incidence or financial burden. Detailing the impact of this disease would further strengthen the significance of this work.

Response: Added accordingly. (see **Lines 2-4 and 5-8, Page 3**).

4. There are a number of typographical and grammatical errors throughout the text. For example:

a. Typos: "benefiting" line 71, page 3; "in details" line 102, page 4; “condensed” line 147 page 6; "MRI plain scanning" line 325, page 13; etc.

b. Grammar / diction errors: The word “charged” should be inserted between “negatively” and “nucleic” in line 75; remove the word “to” prior to “2” on line 147;

the word “living” should be changed to “live” on line 234; “fluorescent signal of PEI/Cy3 did not be detected obviously” should be change to “could not be detected”;“and no fluorescent signal in thoracic aortic arch” lines 236-237 is an incomplete sentence; “were showed” in line 267 should be changed to “are shown”; etc.

Overall, the text should be thoroughly proof read and the grammatical errors corrected.

Response: Thanks for your suggestion. We corrected such errors accordingly. (see **Line 19, Page 3; Line 4, Page 5; Line 19, Page 7; Line 25, Page 15; Line 24, Page 3; Lines 9-10, 12-13 and 14-15, Page 11;** etc.)

5. Articulate and support this statement: "Powerful..." line 74, page 3.

Response: Clarified accordingly. Powerful gene vectors should have good transfection efficiency and low toxicity (see **Lines 21-23, Page 3**).

6. The dose and route of administration should be stated in the paper body, rather than only in the supplement. Dose is typically presented in mg/kg of the RNA. The quantity is low enough to argue for the value of such a therapy, in spite of the frequent dosing required.

Response: Added accordingly in the text. All the mice were treated via angular vein with complexes containing 2.5 nmol miRNA per time. (see **Line 18, Page 12**)

7. The authors state that the “rich hydroxyls can effectively balance the surface potential of complexes and reduce the deleterious effects of excess positive charges” lines 186-187. However, the results shown in Figure 1b show that there are no significant differences in surface charge between the nanomissiles and PEI complexes. This statement is not justified by the data shown and should be removed or altered to accurately reflect the data.

Response: Removed accordingly.

8. Several particle formulations are referred to with -L (like TP-Gd/miRNA-L), which is never identified explicitly. I assume this means it has undergone layer-by-layer self-assembly? The discussion in lines 153-157 comes a little late given the earlier references, and

Response: Thanks for your comments. As mentioned above, layer by layer assembly is deleted. To more clearly describe the nanosystem, “-L” was replaced with ColIV in the revision. TP-Gd/miRNA-ColIV was used instead throughout the work.

9. The bars in Figure 1b surely do not represent the PDI. Should probably at least include the PDI for these four in Table S1.

Response: As suggested, the corresponding PDI was added in **Table S1**.

10. The authors state that the “fluorescent signal of PEI/miR-Cy3 did not be detected obviously” on line 238, however figure 3b does show apparent signal in the PEI/miR-Cy3 group, albeit significantly less than the nanomissile treatment groups.

The text should be modified to accurately address this.

Response: Modified accordingly as suggested. The fluorescent signal of PEI/miR-Cy3 was much lower than those of the TP-involving groups. (see **Lines 14-15, Page 11**).

11. The quantification of delivery in figure 3a is unclear – the y axis and texts mention the relative ROI but do not detail what is actually being measured. Is it the average fluorescence intensity of the ROI that is being compared? Is there any normalization of the fluorescent signal to the area of the ROI? These issues should be clarified in the text.

Response: Clarified accordingly. ROI is relative region of interest which was detected by live imaging system to demonstrate the fluorescence intensity of detected areas. Herein, relative ROI is the average fluorescence intensity with normalization using the miR-Cy3 group. (see **Lines 21-25, Page 11**).

Referee: 2

In this article, Xu et al. have designed nanoparticle TP-Gd and demonstrated its potential in the delivery of microRNA mimetics to... Overall, the data is nicely presented and manuscript is well-written. However, more data are required to confirm the most valuable part of this study—if TP-Gd deliver microRNA “directly” to smooth muscle cells at the site of TAD in animals. Below are my comments:

1. In figure 3b, authors should provide higher magnified pictures to demonstrate that TP-Gd/miR-Cy3-L is indeed “in” the cytoplasm of smooth muscle cells. Current level of magnification can only demonstrate that the TP-Gd/miR-Cy3-L was delivered to TAD region.

Response: Thanks for your kind advice. As suggested, the higher magnified pictures were provided (see **Figure 3b**). SMCs in vessels were stained in green by immunofluorescence staining of α -SMA. Cy3 fluorescence could hardly be detected in the free miR-Cy3 and PEI/miR-Cy3 groups. It was observed that miR-Cy3 was delivered by different TP-containing complexes to SMCs according to the location of α -SMA. (see **Lines 4-8, Page 12**)

2. A followed up question to the first point, what is the organ distribution of TP-Gd/miR-Cy3-L? Author should image whole mouse body to evaluate if TP-Gd/miR-Cy3-L are specifically delivered to TAD, instead of other organs.

Response: As suggested, we performed the experiment of the whole animal biodistribution at different time points of 0-24 h. The corresponding data were added in this revision (see **Figure S8 in Supporting Information**). Although it is difficult to avoid the accumulation of miR-Cy3 in other organs, under the same dosage of administration, the fluorescent signal of thoracic aorta in the TP-Gd/miR-145-CollIV group due to its targeting effect was the strongest among all the groups at each time point. (see **Lines 11-25, Page 11**)

3. In figure 5a, the amount of miR-145 being delivered to SMC at TAD site should be presented as absolute amount (i.e. copy number), instead of relative amount determined by 2ddCt method; otherwise, it is difficult to evaluate the delivery efficacy.

Response: Clarified accordingly. Our delivery efficiency tests of miRNA were based on the earlier works (see **References S3, S4 in Supporting Information (Clin. Cancer Res. 19, 2355-67 (2013); Blood 114, 5331-41 (2009))**), where the comparative cross threshold (2ddCt) method was used to calculate relative quantification of gene expression to test the delivery efficiency of nanoparticle. In future, we will look for cooperation to calculate the absolute amount of delivered miRNA in tissues. In this revision, the references and the test method were addressed. (see **References S3, S4 in Supporting Information (Clin. Cancer Res. 19, 2355-67 (2013); Blood 114, 5331-41 (2009))**; see **Line 25, Page S10 in Supporting Information**)

4. In figure 5c, where exactly on the thoracic arch the KLF4 expression was reduced? Authors should take higher magnified images to demonstrate the IHC result and use arrow or dashed line to label or highlight the site showing high KLF4 in control groups but less KLF4 in miR145 groups.

Response: As suggested, the higher magnified images were provided to demonstrate the IHC result and the arrow was used to label the site. (see **Lines 7-13, Page 14**; see **Figure 5b and Figure S10 in Supporting Information**)

5. In figure 5c, the overall staining of KLF4 looks pretty weak in control group, authors should determine the protein level of KLF4 with western blot to evaluate the effect of miR-145 delivery.

Response: As suggested, in this revision we supplemented the protein quantification by WB for KLF-4 (see **Figure 5c**; see **Lines 15-18, Page 14**). The results of WB were consistent with those of IHC analysis and qRT-PCR.

6. In figure 6b, the labeling of nanoparticle used in control groups should be consistent with figure 6d (i.e, TP-Gd/miRNA should be labeled as TP-Gd/miRNA-neg). It is confusing to see different name for control groups.

Response: Corrected accordingly. (see **Figure 6**; see **Line 23, Page 15**; see **Lines 2, 4, 6, 8, 10, 11, 12, 13, 17, Page 16**)

7. In figure 6d, the signal in the background area (not included in dash lined area) seems to be stronger in negative control group than that in miR-145-L group. To justify the imaging quality, authors should provide lower magnified images for all of the TAD images.

Response: As suggested, lower magnified images for all of the TAD images were provided in this revision (see **Figure S15 in Supporting Information**). To avoid the influences of background signals, the normalized percentage signal enhancements

(%NSE) were calculated by quantitatively analyzing MRI data as described in Supporting Information. (see **Lines 8-18, Page S13 in Supporting Information**)

8. Liver function test should be performed at 24-48 hours after administration of nanoparticle. One hour is too short to detect changes of these common liver and kidney function biomarkers.

Response: Thanks for your suggestion. As suggested, a new liver function test was performed at 48 h. The updated data about blood toxicity assessments were added in this revision. (see **Figure 8, Lines 11-13, Page 17; Lines 22-23, Page S13 in Supporting Information**). All the test groups were similar to those of the blank group without treatment.

9. In figure 7, author should also use lower magnified images for heart, liver and kidney to fairly evaluate the damage or inflammation throughout the organ, if there is any.

Response: As suggested, lower magnified images of major organs were provided in this revision. No obvious pathological lesions were observed in the sections of all the groups. (see **Figure 7 and Figure S16 in Supporting Information; see Lines 9-10, Page 17**)

Referee: 3

The English usage needs some attention.

Response: Thanks for your suggestion. We carefully checked and improved the whole work largely.

The molecular biology was difficult to follow for this clinical reviewer. I am certain that experts in this area will comment on those aspects.

From the clinical point of view, I have the following observations:

1) You seem to have the misconception that TAD (thoracic aortic dissection) is a chronic event. Certainly, the aortic wall does deteriorate over time. However, TAD is an INSTANTANEOUS event that occurs in a vulnerable aorta. Please review your entire manuscript, especially the clinical observations, with this in mind.

Response: Based on your kind advice, the information about TAD was clarified in this revision. Herein, in our manuscript, what we mainly focus on is the process before TAD (pre-TAD). In pre-TAD, this process may be a chronic event and needs to be detected and treated. (see **Lines 5-6, Page 3; Lines 3-7, Page 15**)

2) You seem to have the misconception that it is difficult to diagnose TAD. (You say: "diagnostic methods cannot clearly demonstrate the location and size of TAD.[50-55]". That is completely untrue. Cardiac echocardiography, computed tomography, and magnetic resonance imaging are all capable of detecting TAD with a very high level of accuracy.

Response: Thanks for your comments. Herein, what we mainly focus on is the process before TAD (pre-TAD). We corrected the corresponding description about the

diagnostic methods as below. Clinical diagnosis of TAD relies on medical imaging methods, such as MRI, transthoracic echocardiography (TTE) and computed tomography angiography (CTA).^[50,51] However, these methods cannot provide early detection of dissection before intimal tear (the process of pre-TAD). (see **Lines 3-7, Page 15**).

3) You seem to have the misconception that TAD is clinically silent (You say:"TAD develops without obvious clinical manifestation, deteriorates quickly and finally lead to dissection rupture and organ ischemia.) TAD has very specific and dramatic clinical manifestations (chest or back pain, organ ischemia).

Response: Thanks for your comments. Herein, what we want to show is that pre-TAD was without obvious clinical manifestation. The information about TAD was clarified in this revision. (see **Lines 5-6, Page 3**)

4) Your references do not always seem pertinent where you have cited them. Please check every one.

Response: Checked carefully thorough the work. The unsuitable references were changed. (see **References 3, 34, 40, 42, 50-54 and References S3, S4, S6, S7 in Supporting Information**)

Congratulations on your very promising findings, which may hold great clinical importance.

Response: Thank you very much for your encouragement.

Now we are submitting our revised manuscript along with the electronic supplementary information. We are looking forward to hearing from you soon.

Sincerely yours,

Dr. Xu

Reviewers' Comments:

Reviewer #1:

Remarks to the Author:

The authors sufficiently responded to my prior comments.

Reviewer #2:

Remarks to the Author:

Overall, the new data is not convincing to demonstrate that the particle is specifically targeted to aorta SMC in vivo. Also, it is questionable that the amount of miR being delivered is high. The particle do deliver miR to aorta and ameliorate TAD in the animal model, but author should avoid using "precise targeting" or "specific to aorta" to describe the delivery property of the particle. Please revise that unless there is new evidences to clarify the issue.

Please find the point-to-point new comments below:

1. Acceptable. The evidence is not very strong due to the still lower magnification but it seems like the particle is in the cells because not much intercellular space in this tissue region.

2. Not acceptable. The distribution of particle in the kidney is stronger than that in aorta. In fact, the particle is delivered to all organs being imaged except spleen. However, it is totally possible that the particle is only trapped in these organs but not delivered successfully into the cytoplasm. That is why a highly magnified and high-resolution image is so critical to justify if the particle is specifically delivered to the cells in aorta, instead of other organs.

3. Not acceptable. The issue with comparative 2ddCt method is that delivery could be very poor but the fold change looks mistakenly good. For example, although the TP-Gd-miR-145-CoIV group has 4-fold delivery efficiency compared to the control, the raw Ct could be 34 vs 36 between these two groups; if that is the case, then the 4-fold increase does not make any impact to the cells. If it is technically difficult to detect the absolute miR copy number, authors should provide raw Ct for all groups and add an extra one for housekeeping genes such as Rnu6b, which is normally between a Ct of 20~24 with 100ng RNA, as a comparison.

4. Need to be improved. It is hard to believe those arrowed spots are positive cells because the color is light brown instead of a typical dark brown color for immunohistochemistry using 3,3'-diaminobenzidine. The light brown color is usually background due to intracellular peroxidase or the antibody gone bad.

5. Acceptable.

6. Acceptable.

7. Acceptable.

8. Acceptable.

9. Need to be improved. The data looks like all organs are not damaged, but the lower magnified image is too small for reader to determine the fact. Consider the cost for publishing so much information, please enlarge these images to a reasonable size and incorporate into the supporting information.

Reviewer #3:

None

REVIEWER REPORT(S):

Referee: 1

The authors sufficiently responded to my prior comments.

Response: Thank you very much for your encouragement.

Referee: 2

Overall, the new data is not convincing to demonstrate that the particle is specifically targeted to aorta SMC in vivo. Also, it is questionable that the amount of miR being delivered is high. The particle do deliver miR to aorta and ameliorate TAD in the animal model, but author should avoid using "precise targeting" or "specific to aorta" to describe the delivery property of the particle. Please revise that unless there is new evidences to clarify the issue.

Response: Thank you very much for your comments. As suggested, in this new revision, the words 'precise' and 'specific' were deleted. (see **Title; Line 14, Line 22, Page 2; Line 14, Line 20, Page 4 and Lines 13-14, Page 18**)

1. Acceptable. The evidence is not very strong due to the still lower magnification but it seems like the particle is in the cells because not much intercellular space in this tissue region.

Response: We appreciated for your agreements.

2. Not acceptable. The distribution of particle in the kidney is stronger than that in aorta. In fact, the particle is delivered to all organs being imaged except spleen. However, it is totally possible that the particle is only trapped in these organs but not delivered successfully into the cytoplasm. That is why a highly magnified and high-resolution image is so critical to justify if the particle is specifically delivered to the cells in aorta, instead of other organs.

Response: Thank you very much for your comments. We agreed with you that these nanoparticles might only trapped in some organs. Although it is difficult to avoid the accumulation of miR-Cy3 in other organs, comparing with the test groups without ColIV, the fluorescent signal of thoracic aorta in the TP-Gd/miR-145-ColIV group was the strongest at each time point (see **Lines 21-22, Page 11 and Figure S8 in Supporting Information**). Such data supported that TP-Gd/miRNA-ColIV could target to SMC of injured aorta, probably not specifically to aorta. As suggested above, in this new revision, words 'precise' and 'specific' were deleted. (see **Title; Line 14, Line 22, Page 2; Line 14, Line 20, Page 4 and Lines 13-14, Page 18**)

3. Not acceptable. The issue with comparative 2ddCt method is that delivery could be very poor but the fold change looks mistakenly good. For example, although the TP-Gd-miR-145-ColIV group has 4-fold delivery efficiency compared to the control, the raw Ct could be 34 vs 36 between these two groups; if that is the case, then the 4-fold increase does not make any impact to the cells. If it is technically difficult to detect the absolute miR copy number, authors should provide raw Ct for all groups and add an extra one for housekeeping genes such as Rnu6b, which is normally

between a Ct of 20~24 with 100ng RNA, as a comparison.

Response: Thank you very much for your comments. As suggested, we provided all the raw Ct values of miR-145 and U6 for all the groups (see **Table S3 in Supporting Information**). All the average Ct values of miR-145 in different groups ranged from 19 to 24 which demonstrated the high reliability of this method (see **Lines 12-14, Page S11 in Supporting Information**).

4. Need to be improved. It is hard to believe those arrowed spots are positive cells because the color is light brown instead of a typical dark brown color for immunohistochemistry using 3,3'-diaminobenzidine. The light brown color is usually background due to intracellular peroxidase or the antibody gone bad.

Response: Thank you very much for your suggestions. Based on your comment, we provided clearer images and re-drew the arrows to show KLF4 expression. (see **Figure 5b and Figure S10 in Supporting Information**)

5. Acceptable.

Response: We appreciated for your agreements.

6. Acceptable.

Response: We appreciated for your agreements.

7. Acceptable.

Response: We appreciated for your agreements.

8. Acceptable.

Response: We appreciated for your agreements.

9. Need to be improved. The data looks like all organs are not damaged, but the lower magnified image is too small for reader to determine the fact. Consider the cost for publishing so much information, please enlarge these images to a reasonable size and incorporate into the supporting information.

Response: As suggested, all the lower magnified images were enlarged to a reasonable size and were moved into Supporting Information. (see **Figure 7 and Figures S16, S17 in Supporting Information**)

Reviewers' Comments:

Reviewer #2:

Remarks to the Author:

The authors sufficiently improved the data and truthfully demonstrated the nature of this nanoparticle. All changes are accepted.

Referee: 2 The authors sufficiently improved the data and truthfully demonstrated the nature of this nanoparticle. All changes are accepted. Response: Thank you very much for your encouragement.